# Global metabolic interaction network of the human gut microbiota for context-specific community-scale analysis

Jaeyun Sung[1,2,3], Seunghyeon Kim[1,4,5], Josephine Jill T. Cabatbat[1], Sungho Jang[6], Yong-Su Jin[7,8], Gyoo Yeol Jung[6,9], Nicholas Chia[10,11,12] & Pan-Jun Kim[1,4,13]

A system-level framework of complex microbe–microbe and host–microbe chemical cross-talk would help elucidate the role of our gut microbiota in health and disease. Here we report a literature-curated interspecies network of the human gut microbiota, called NJS16. This is an extensive data resource composed of $\sim$570 microbial species and 3 human cell types metabolically interacting through $>4{,}400$ small-molecule transport and macromolecule degradation events. Based on the contents of our network, we develop a mathematical approach to elucidate representative microbial and metabolic features of the gut microbial community in a given population, such as a disease cohort. Applying this strategy to microbiome data from type 2 diabetes patients reveals a context-specific infrastructure of the gut microbial ecosystem, core microbial entities with large metabolic influence, and frequently produced metabolic compounds that might indicate relevant community metabolic processes. Our network presents a foundation towards integrative investigations of community-scale microbial activities within the human gut.

[1] Asia Pacific Center for Theoretical Physics, Pohang, Gyeongbuk 37673, Republic of Korea. [2] Center for Computational and Integrative Biology, Massachusetts General Hospital and Harvard Medical School, Boston, Massachusetts 02114, USA. [3] Broad Institute of MIT and Harvard, Cambridge, Massachusetts 02142, USA. [4] Department of Physics, Pohang University of Science and Technology, Pohang, Gyeongbuk 37673, Republic of Korea. [5] The Abdus Salam International Centre for Theoretical Physics, 34151 Trieste, Italy. [6] Department of Chemical Engineering, Pohang University of Science and Technology, Pohang, Gyeongbuk 37673, Republic of Korea. [7] Department of Food Science and Human Nutrition, University of Illinois at Urbana-Champaign, Urbana, Illinois 61801, USA. [8] Carl R. Woese Institute for Genomic Biology, University of Illinois at Urbana-Champaign, Urbana, Illinois 61801, USA. [9] School of Interdisciplinary Bioscience and Bioengineering, Pohang University of Science and Technology, Pohang, Gyeongbuk 37673, Republic of Korea. [10] Microbiome Program, Center for Individualized Medicine, Mayo Clinic, Rochester, Minnesota 55905, USA. [11] Department of Surgery, Mayo Clinic, Rochester, Minnesota 55905, USA. [12] Department of Biomedical Engineering, Mayo College, Rochester, Minnesota 55905, USA. [13] Department of Physics, Korea Advanced Institute of Science and Technology, 291 Daehak-ro, Yuseong-gu, Daejeon 34141, Republic of Korea. Correspondence and requests for materials should be addressed to P.-J.K. (email: pjkim@kaist.ac.kr).

The microbial habitat within the human intestine is the site of an extraordinarily complex and dynamic symbiosis. Central to the structure and evolution of the resident gut microbial community (gut microbiota) are the various interactions between microbes and with their chemical environment[1,2]. Colonic microbes survive and grow by consuming diet-derived and host-derived chemical compounds, as well as metabolic byproducts excreted by other microbes[3]. Undigested dietary macromolecules and host-derived substrates (such as mucin) are broken down by microbial species and then the solubilized molecules become available to other members of the community as public goods for uptake. Furthermore, inherent microbial activities involving the import of metabolic nutrients and export of metabolic byproducts give rise to both competition for resources and cooperative relationships, such as metabolic cross-feeding, among resident microorganisms in the gut environment[4]. In addition, the interactions of the gut microbiota with the host are increasingly recognized to have an impact on many aspects of human health and disease[5,6]. For example, microbial fermentation products such as short-chain fatty acids (SCFAs) have active roles in normal host physiology, as energy sources for colonocytes, regulators of gene expression and cell differentiation, and anti-inflammatory agents[7,8]. On the other hand, some metabolic byproducts can be toxic, impairing host tissue function and promoting the onset and progression of disease[9]. Taken together, numerous microbe–microbe and microbe–host interconnections serve as the basis of a complex ecological network in the human gut.

Recent advances in sequencing technologies and metagenomics have revealed associations between the abundance of taxonomic groups (or their genetic repertoire) and a number of disorders, including obesity, inflammatory bowel disease, colorectal cancer and type 2 diabetes (T2D)[10–13]. Such descriptive, profiling investigations offer important insights into taxonomic and functional variations relevant to host phenotypes; yet, a mechanistic and comprehensive understanding of those observed results remains elusive. Notwithstanding the importance of individual microbial species, the consequent impact of the microbiota on the host would be largely attributed to the collective activities of numerous microbial species and metabolic compounds, thoroughly interlinked by network relationships. This realization calls for an integrative network-based approach for a system-level understanding of the human gut microbiota[14,15]. If available, a comprehensive map of molecular interactions between microbial species could be used to integrate the vast collection of previous findings into a global network context.

In the microbiome research field, a common practice to build a microbial interaction network has been based on statistical correlations of taxa abundances across samples[16,17]. However, correlation-based inference networks hardly provide explicit mechanistic details behind the identified correlations. Another existing approach is to map the entire metabolic pathways by the direct annotation of metagenome sequences[18,19]. Despite the advantage of evaluating the metabolic potential of the community in its entirety, this method does not segregate biochemical reactions to those of different species, preventing its application for the analysis of interspecies interactions. Furthermore, based on the information of different metabolic networks of individual microbial species, there are previous works that have modelled diverse interspecies interactions explicitly mediated by imported or exported metabolites[20,21]. Yet, these works have relied on error-prone automated identification of transportable (importable or exportable) metabolites and therefore are possibly incomplete or inaccurate to some extent. Despite ongoing computational efforts to describe microbial interactions using biologically realistic (manually curated constraint-based or simplified kinetic) metabolic models[22–24], most of these models have not yet reached the scale of diversity in the gut community, which typically comprises hundreds of different microbial species (it is noteworthy that this scale of microbial diversity has been recently covered by semi-automatically generated, constraint-based metabolic models[25]).

Here we present a global interspecies metabolic interaction network of the human gut microbiota, NJS16. The information upon which the network architecture stands is primarily from literature-based annotations. Therefore, our network maps the landscape of existing biological knowledge and curated experimental data. To demonstrate the utility of our network, we developed a mathematical framework for analysing gut microbial communities in a given population, such as a cohort of T2D patients. Recent studies have indicated that, as a prominent environmental factor, alterations in the gut microbiota contribute to the pathology of this disease[12,26]. Combined with faecal metagenomic information from T2D patients[12], our network analysis reveals a community-scale infrastructure of metabolic influence within the T2D gut ecosystem.

## Results

**Metabolite transport network of the human gut microbiota.** We aimed to construct a community-level network composed of microbial species populating the human gut (Fig. 1). To construct our network, we started by applying a phylogenetic analysis tool on previously published, shotgun metagenomic sequencing data from faecal samples of Chinese individuals[12]. Further details of the microbiome data and of the taxonomic profiling method are provided in Methods and Supplementary Data 1. Next, we extensively searched the published literature for all annotated, mainly experimental, information that describes the small-molecule metabolites (for example, monosaccharides, disaccharides, SCFAs, vitamins and gases), which are transported into, and/or out of, the microbial species identified in the microbiome samples (see Supplementary Data 2 for bibliography of literature references). To complement our list of annotated microbe–metabolite associations, we included macromolecule degradation reactions that involve microbes and the macromolecules (for example, cellulose, hemicellulose, inulin, starch and mucin) which the microbes are known to degrade, as well as the resulting degradation products (for example, D-glucose and cellobiose from cellulose, N-acetylglucosamine, N-acetylneuraminate, L-fucose and sulfate from mucin). Although tissue cells of the human host are not physically a part of the gut microbiota, in this study we view them as a functional extension of the bacterial and archaeal community residing in the colon, because host cells either can directly affect or can be affected by microbial metabolism. The specific host cells that we considered were the colonocyte (cell-type-specific metabolic model from Recon 2)[27], the goblet cell (for mucin secretion) and the hepatocyte (for glycine- or taurine-conjugated bile acid export). Lastly, we linked all microbes and host cells to their associated metabolites and macromolecules into a comprehensive reference map of human gut microbiota and chemical compound relationships. More specifically, a community member and a chemical compound are then connected by a (directed) link if the organism can import and/or export the metabolite, or degrade the macromolecule (see Methods and Fig. 2 for further details of our network construction approach and the assessment of its possible biases, respectively).

We present NJS16, a literature-curated community-level network of the human gut microbiota organized through metabolite transport (Fig. 1). Our network is a compilation of

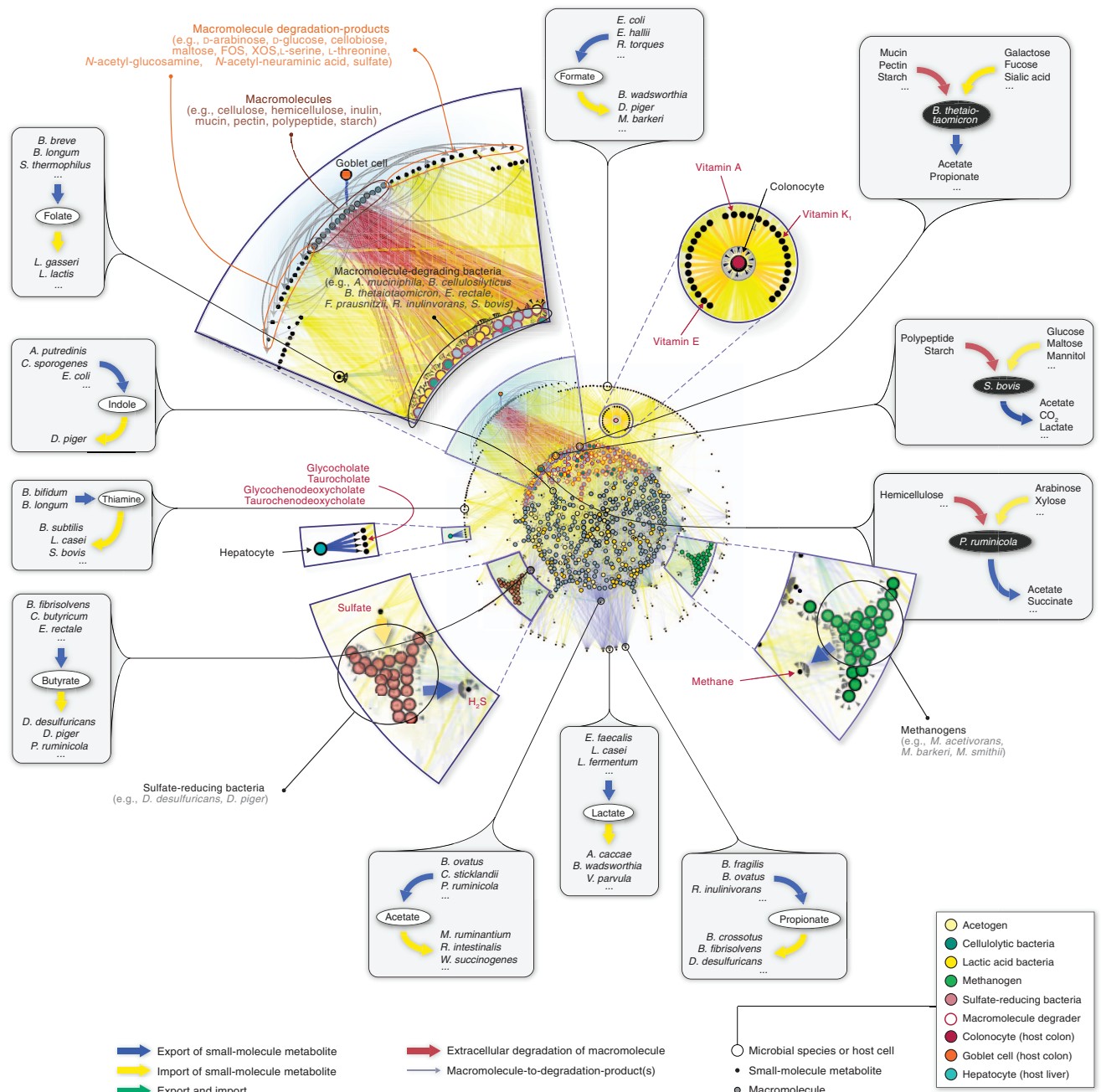

**Figure 1 | Global landscape of the human gut microbiota organized through metabolite transport.** Overview of NJS16. The import of nutrients (yellow arrows) and export of metabolic byproducts (blue arrows) comprise the organizational basis of the gut microbial community. Microbes of common metabolic function are clustered together as functionally similar groups (large coloured nodes). Within the microbial community, competition exists for the consumption of the same metabolites (small black nodes); cooperative relationships also occur, in the form of (i) interspecies cross-feeding, as exemplified in the figure insets, and (ii) macromolecule degradation, wherein a microbe degrades macromolecules in its extracellular space (red arrows), thereby releasing degradation products (grey arrows stemming from macromolecule nodes) as public goods. As a functional extension of the bacterial and archaeal community residing in the colon, human host cells either can directly affect, or can be affected by, microbial metabolism. Host cell types in the network are: (i) colonocytes, which absorb nutrients produced by certain microbes, (ii) goblet cells, which secrete complex mucus glycoproteins for mucin-degrading microbes, and (iii) hepatocytes, which, although not part of colonic tissue, secrete conjugated bile acids that are consumed by microbes.

4,483 annotated transport or degradation reactions (from about 400 research articles, reviews and textbooks) between 244 metabolic compounds (229 small molecules and 15 macromolecules) and 570 microbial species and human cell types (511 bacteria, 56 archaea and 3 host cells) (see Supplementary Data 2 for information on all nodes and arrows in NJS16, on the associations between macromolecules and their breakdown products, and on which microbial species have been previously well studied regarding their relationship to the human gut). To investigate the inherent network topology of NJS16, we calculated the number of metabolites imported or exported by each microbial species. Each species in NJS16 imports 5.1 and exports

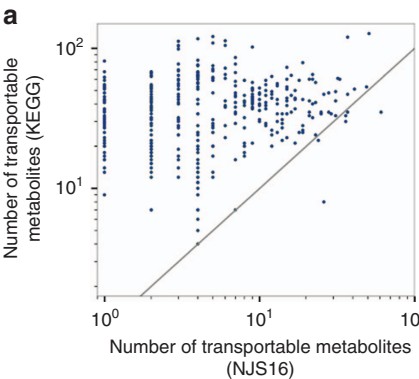
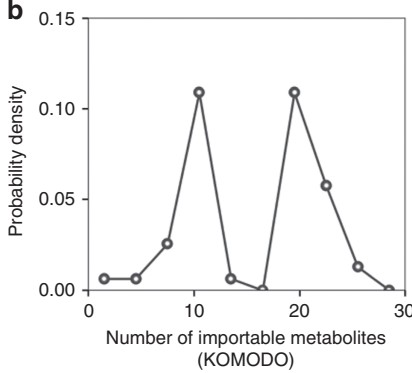

**Figure 2 | Comparison of transportable metabolites from NJS16 and those inferred from existing databases.** (**a**) For each microbial species (each data point), the horizontal and vertical axes represent the number of its transportable metabolites from NJS16 and that inferred from the Kyoto Encyclopedia of Genes and Genomes (KEGG)[69], respectively. The latter was obtained by counting the species' KEGG compounds that are common to any of the entire transportable metabolites in NJS16. This KEGG-based estimation would result in many false positives; however, the KEGG information might be relatively free of the literature bias that NJS16 harbours and thus can possibly serve as a more unbiased counterpart to NJS16. Most data points are located over the grey diagonal, indicating that most species have more transportable metabolites according to KEGG than to NJS16. The presence of species with few metabolites in NJS16 can be, at least in part, attributed to literature bias with false negatives in the species' metabolites. (**b**) The vertical axis represents the distribution of the probability $P(k)$ that a given microbial species has $k$ metabolites (horizontal axis) in its defined growth media whose information is from the Known Media Database (KOMODO)[70]. We considered common species between NJS16 and KOMODO (only when found to have defined media information) and counted their media components that are common to any of the transportable metabolites in NJS16. For a given species, the number of such media components may possibly approximate the number of its importable metabolites. Compared with Fig. 3a, which is derived from NJS16, **b** exhibits peaks at large metabolite numbers on the horizontal axis, possibly indicating false negatives in NJS16's importable metabolites. Yet, given KOMODO's low coverage of microbial species in NJS16 (9.2%) and given that microbes may not necessarily import all compounds in their defined growth media, our results warrant a cautious interpretation.

3.9 metabolites on average (median 3 metabolites for both cases), and the probability that a given species imports (or exports) $k$ metabolites follows an exponential distribution $P(k) \propto e^{-rk}$ ($r \approx 0.2$ and $0.4$ for the import and export cases, respectively; see Fig. 3a,b). The most promiscuous species is *Bacteroides thetaiotaomicron*, which imports 34 and exports 29 metabolites. Conversely, for each metabolite, we calculated the number of species importing or exporting that metabolite. The probability that a given metabolite is exported by $k$ species follows a power-law distribution $P(k) \propto k^{-\gamma}$ ($\gamma \approx 1.6$), which is much broader than the previous exponential fits; such a broad distribution is also observed from imported metabolites (Fig. 3c,d). Specifically, glucose and acetate are the most frequent substrate and product, respectively, and are imported by 118 (20.8% of the total species) and exported by 251 species (44.3% of the total species). In contrast, an average metabolite is imported by 13.4 and exported by 20.7 species (median 7 and 4 species, respectively). Collectively, these results indicate that metabolites are highly uneven in terms of their use by species.

Microbes compete against each other for the utilization of available substrates (for example, carbon, nitrogen and phosphorus sources) in the human gut. In our network, this competition is especially a common feature among groups of microbes that share common metabolic and physiological characteristics, such as acetogens and sulfate-reducing bacteria (interestingly, our network and microbiome data show that the similarity in two species' nutritional profiles is positively correlated with the species' co-occurrence ($\rho = 0.29$ and $P = 0.02$), in agreement with a previous claim[20] when the same measures were applied). Cooperative relationships are also present, in the form of (i) interspecies cross-feeding, wherein a metabolic byproduct of one microbe is a nutrient of another (see each Fig. 1 inset that shows microbes surrounding a particular metabolite) and (ii) macromolecule degradation, wherein a microbe, via its ability to degrade macromolecules and thereby release the degradation products into the

microenvironment as public goods, provides nutrients not only for itself, but also for other community members. For example, lactate produced by *Bifidobacterium*, *Enterococcus* and *Lactobacillus* species is imported by lactate consumers such as *Anaerostipes caccae* and *Bilophila wadsworthia*. Macromolecule degraders, such as those that target hemicellulose (for example, *Prevotella ruminicola*), can provide the degradation products (for example, D-arabinose, D-galactose and xylooligosaccharides) to themselves, as well as to nearby microbes.

As such, host cells are also involved in the cooperative metabolic relationships within the gut microbiota: colonocytes can absorb metabolites produced by microbes (for example, SCFAs, amino acids and vitamins); goblet cells secrete complex mucus glycoproteins as part of the intestinal mucosal layer, which are a target for mucin-degrading microbes (for example, *Akkermansia muciniphila* and *B. thetaiotaomicron*); and hepatocytes export glycine- or taurine-conjugated bile acids, which eventually flow into the intestine and are taken up by bile-acid-consuming microbes (for example, *Bacteroides fragilis* and *Clostridium perfringens*). As clearly seen in these examples, metabolite-driven cooperative relationships are pervasively seen throughout the entire network (Fig. 1). Using NJS16, we can now begin to explore the organizing characteristics of global microbial metabolic activities in the human gut environment.

**Context-specific microbial MIN.** Herein, we demonstrate the potential of our global metabolite transport network of the gut microbiota for elucidating context-specific, community-level features. We developed a mathematical framework and applied it on the aforementioned Chinese microbiome samples, which were previously obtained from T2D patients and non-diabetic controls[12]. In particular, we undertook the analysis of these data with an aim to identify the most representative microbial and metabolic features of T2D gut microbiota, and to gain insight into relevant microbe–microbe and microbe–host relationships.

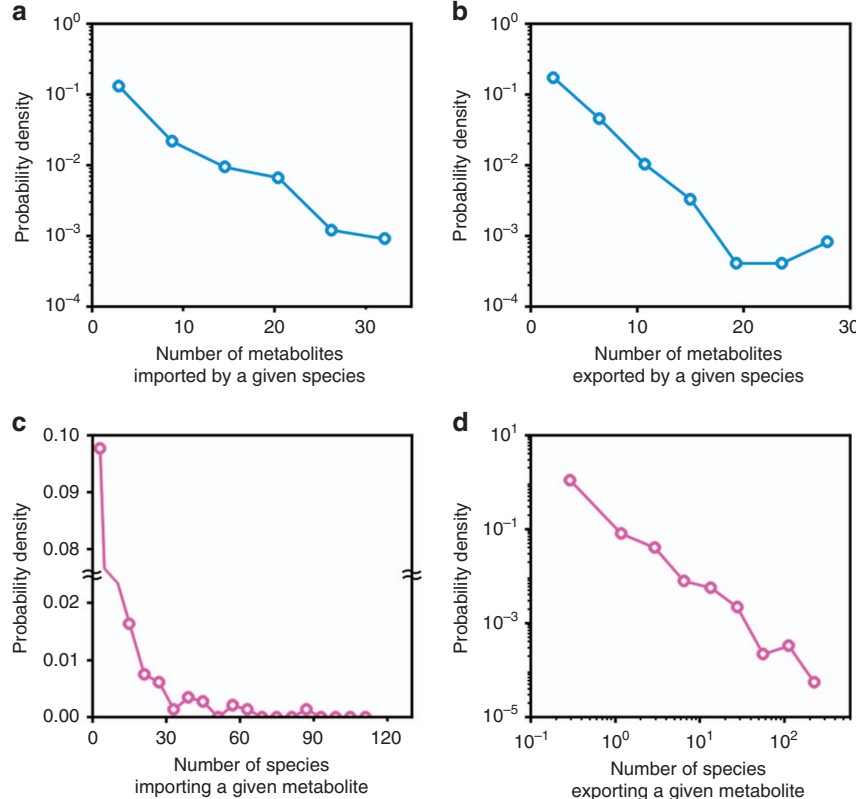

**Figure 3 | Network structural properties of NJS16.** In **a,b**, the vertical axis represents the distribution of the probability $P(k)$ that a given microbial species imports (**a**) or exports (**b**) $k$ metabolites on the horizontal axis. In **c,d**, the vertical axis represents the distribution of the probability $P(k)$ that a given metabolite is imported (**c**) or exported (**d**) by $k$ species on the horizontal axis. (**a,b**) Exponential distributions $P(k) \propto e^{-rk}$ with $r \approx 0.2$ for **a** and $r \approx 0.4$ for **b**. (**c,d**) More right-skewed distributions than **a,b**. (**d**) A power-law distribution $P(k) \propto k^{-\gamma}$ with $\gamma \approx 1.6$.

Recent studies have shown that gut microbiota composition and its functional traits can vary according to socio-demographic and environmental factors, such as ethnicity, gender, age and diet[28–30]. These factors can also have a profound impact towards diseases such as T2D[31]. This indicates that comparative microbiome analyses (for example, case versus control) should be conducted in well-characterized cohort populations controlled for sources of confounding variation. Therefore, for subsequent analyses, we selected T2D and non-diabetic control microbiome samples from a demographic cohort characterized as male, mid-age and normal weight. This particular cohort was chosen, because it has the largest sample size among all cohorts, as well as comparable sample numbers in both phenotypes (control $n = 21$; T2D $n = 11$). Because of a possible confounding effect on the gut microbiome by an oral anti-diabetic medication[32], we used T2D samples from only metformin-untreated patients (Methods). For this male, mid-age and normal weight cohort, we then found microbial entities differentially abundant or scarce in T2D (Methods). Here, a microbial entity represents a single microbial species or a group of multiple microbial species; a group pertains to microbial species of either a genus or a metabolic clique, which is defined here as a group of species that import, export or degrade the same metabolite or macromolecule (for example, glucose importers, butyrate exporters or cellulose degraders). A total list of relevant microbial entities is presented in Supplementary Data 3.

Next, we applied NJS16 as a reference map to build the community structure of microbial entities abundant or scarce in T2D, and to understand how they metabolically influence each other. This network, which we will call henceforth the microbial metabolic influence network (MIN), is based on actual microbial relative abundance information from the relevant T2D and control microbiome samples, that is, context-specific abundance information. A brief description of how we can construct these context-specific microbial MINs is as follows: in complex, microbial ecosystems, a microbial entity can provide nutrients to another entity via interspecies cross-feeding of metabolic byproducts and/or release of macromolecule degradation products. This positive impact may potentially promote microbial growth. In contrast, a microbial entity can limit another entity's access to nutrients via competition for the same metabolites. This negative impact may potentially inhibit microbial growth. Based on the combination of such positive and negative effects, we can leverage information from NJS16 to formulate and quantify the net metabolic influence of a microbial entity $i$ on another entity $j$ ($W_{ij}$). If each entity $i$ or $j$ is a single species but not a group of multiple species, $W_{ij}$ is estimated as

$$W_{ij} \approx \sum_{k}^{g_{jk}^c=1} \left\{ \alpha \frac{n_i g_{ik}^p}{\sum_m n_m g_{mk}^p} - \frac{n_i g_{ik}^c}{\sum_m n_m g_{mk}^c} \right\},$$

where $n_i$ characterizes entity $i$'s abundance, $k$ denotes the metabolites consumed by entity $j$, $g_{ik}^{p(c)} = 1$ if entity $i$ produces (consumes) metabolite $k$, or otherwise, $g_{ik}^{p(c)} = 0$, and $\alpha$ is a constant. Full details of $W_{ij}$, including its formulation, the incorporation of macromolecule degradation, extension to the case where entity $i$ or $j$ is a multi-species group and the determination of $\alpha$, are described in Methods and Supplementary Methods. Overall, if a microbial entity $i$ is found to have a higher growth-promoting capability than a growth-inhibiting capability

towards another microbial entity $j$, then $W_{ij} > 0$, and we classify this interaction as having a net positive metabolic influence; if there is a higher growth-inhibiting capability than a growth-promoting capability, then $W_{ij} < 0$, and we classify this interaction as having a net negative metabolic influence. This approach allows us to construct a community-level network of positive and negative metabolic influences between pairs of microbial entities differentially abundant or scarce in T2D (codes are available in Supplementary Software). Furthermore, we include in these networks cross-feeding interactions between microbes and the host (Methods and Supplementary Methods), along with their corresponding metabolites.

In Fig. 4, we show the MIN composed of microbial entities associated with a male, mid-age and normal weight cohort. It features the interplay of positive and negative metabolic influences among 125 microbial entities. 116 of these 125 are differentially abundant in T2D compared with control, whereas 9 are differentially scarce. As mentioned above, one major way a microbial entity exerts a net positive metabolic influence is through its ability to release macromolecule degradation products for consumers of those public goods. For instance, *Acidothermus cellulolyticus* can degrade cellulose in this MIN. Cellulose degradation supplies D-glucose, cellobiose and cellulose oligosaccharides to the microbial community. As another example, *A.*

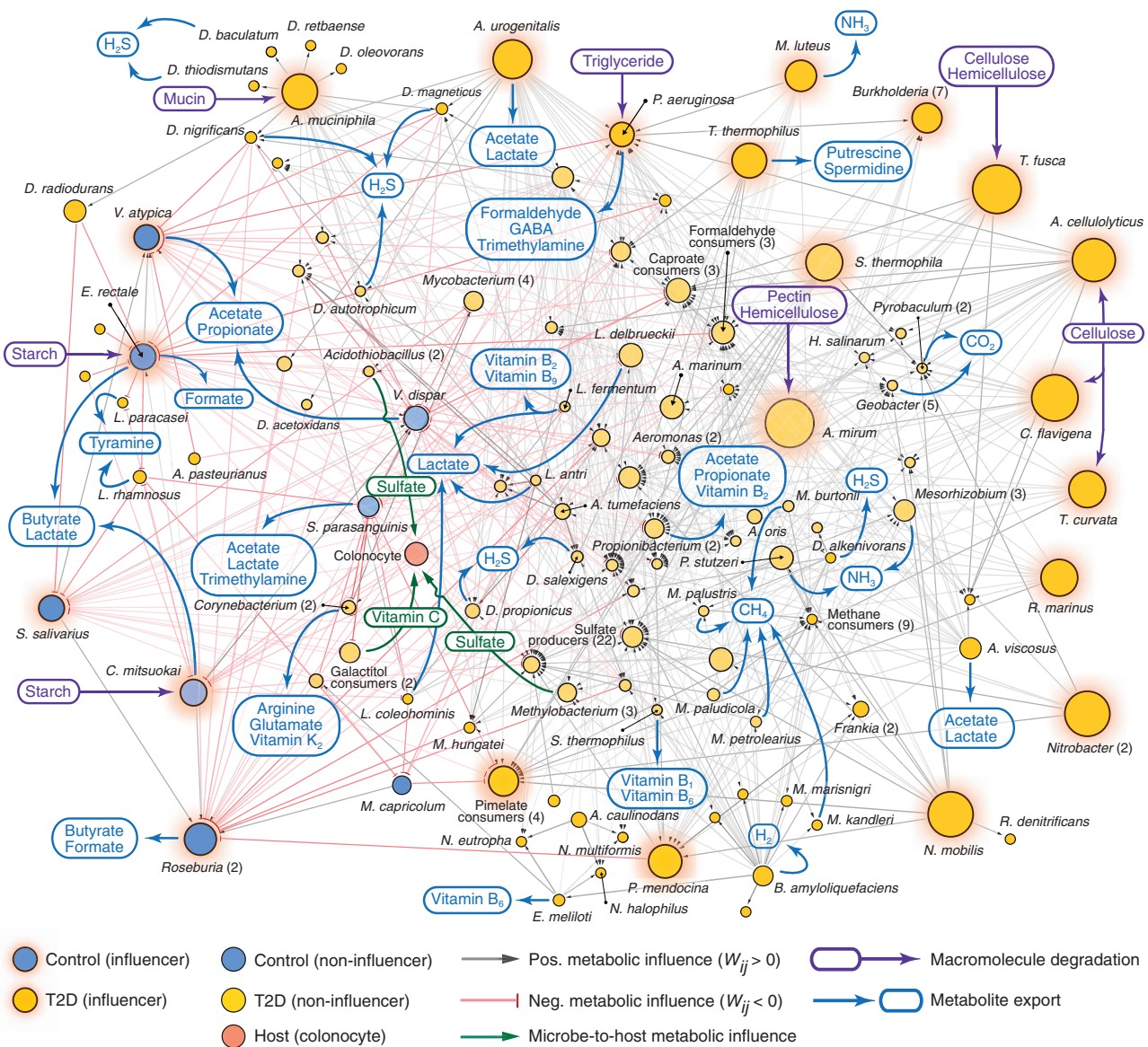

**Figure 4 | MIN among the most representative microbial entities of T2D in a given demographic cohort.** We identified community-wide metabolic influence relationships between microbial entities differentially abundant in T2D (gold nodes) and in non-diabetic control (blue nodes) in a male, mid-age and normal-weight cohort. Specifically, the network is characterized by positive (grey arrow) and negative (red arrow) metabolic influences between pairs of microbial entities. Furthermore, entities that are highly influential—by exerting a metabolic influence towards a substantial number of microbial entities—are depicted as network influencers (nodes with orange background. See also Fig. 5a). The number of species that compose each microbial group is shown in parentheses next to the respective group's name. Microbe-to-host (colonocyte) cross-feeding relationships that were predicted to be of representative importance are shown in green arrows, with some examples of the corresponding metabolites. Macromolecule degradation by individual microbial species, or by multiple species in a microbial group, is exemplified through purple arrows. Full lists of microbial entities and compounds in the networks are too dense for direct visualization and therefore only a part of them are presented. Full details of this network are available in Supplementary Data 3.

*muciniphila* in MIN participates in mucin degradation, providing the degradation products to the microbial community. The other major way a microbial entity can have a net positive metabolic influence is through cross-feeding of exported metabolites. For example, a member of the *Propionibacterium* genus produces riboflavin for *Lactobacillus delbrueckii* in the MIN.

Apart from the positive metabolic influence relationships examined above, the MIN also includes links of net negative metabolic influence. Primarily, this type of relationship indicates possible competition for the same resources, which may lead to growth suppression of a microbial entity by another. *Catenibacterium mitsuokai*, *Eubacterium rectale* and *Veillonella atypica*, which were all found to be scarce in T2D (conversely, abundant in control), have net negative metabolic influence to many microbial entities found to be abundant in T2D (conversely, scarce in control). For example, in the case of *E. rectale*, there is possible competition with *L. delbrueckii* and *Lactobacillus fermentum* for lactose consumption. As an overview of the MIN from a male, mid-age, and normal weight cohort, Supplementary Data 3 provides descriptions of which metabolites determine each metabolic influence between pairs of microbial entities.

Taken together, our MIN of microbial entities portrays a roadmap of how chemical interactions come into effect for community members in the gut environment. Given that microbial organisms in the gut can synthesize and export various chemical compounds, these microbial products can be seen as potential modulators of microbe–host interactions. Although a thorough examination of all possible metabolite production is necessary towards understanding global microbe–host interactions, a more effective means would be to focus our efforts on the strongest and most relevant metabolic interactions. For this purpose, we devised a computational method to pinpoint the chemical compounds and their associated microbial entities that compose the most representative interactions with the host (Methods; full details of our pipeline are presented in Supplementary Methods). A list of these microbe–host relationships is provided in Supplementary Data 3.

In contrast to the entire spectrum of microbe–metabolite associations presented in Fig. 1, our T2D-associated MIN illustrated in Fig. 4 offers a focused view of the most relevant microbe–microbe and microbe–host metabolic relationships. These interactions could be interesting starting points for further interrogations on context-specific gut microbiota at the community level.

**Hierarchy of the microbial MIN.** In the influence network shown in Fig. 4, the observation of hubs (which are microbial entities that exert a metabolic influence to a relatively high

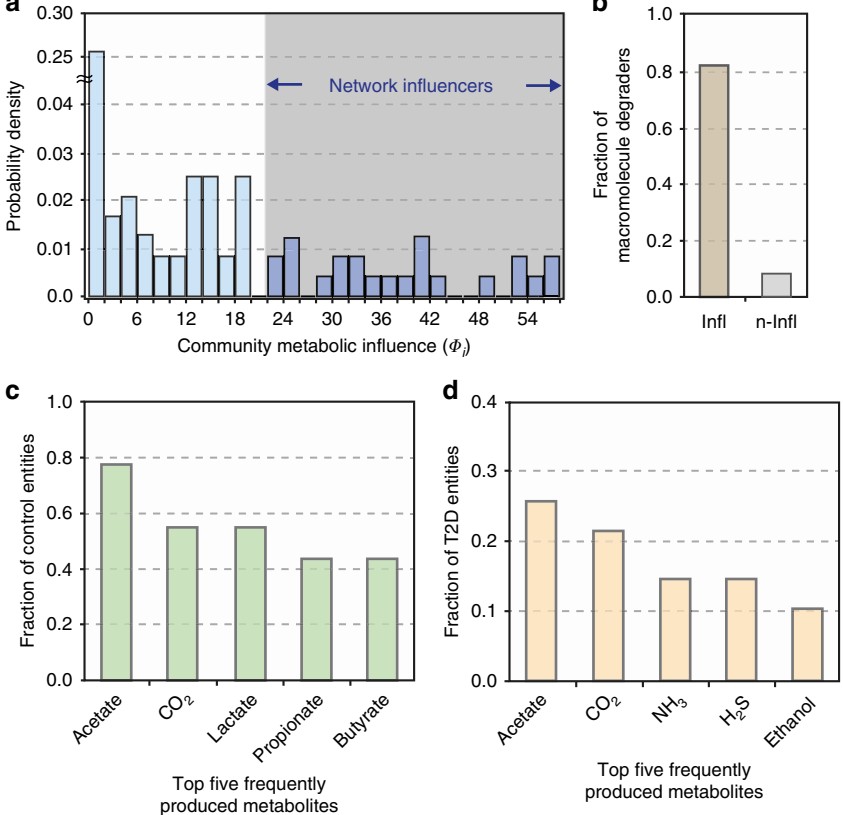

**Figure 5 | Characteristics of the MIN.** The community metabolic influence ($\Phi_i$) of each microbial entity $i$ was found for all nodes of the influence network (Fig. 4) and its probability distribution is shown in **a**. This plot was used to identify microbial entities with relatively large metabolic influences; specifically, a transition point was chosen after a drop-off in the probability density of community influences (Methods). Microbial species or groups with relatively large metabolic influence (shaded region) were designated as network influencers. (**b**) Network influencers (denoted by 'Infl') have a higher proportion of macromolecule degraders than non-influencers (denoted by 'n-Infl'), showing that macromolecule degradation is one of their key hallmarks. (**c**) Top five most frequently produced metabolites by microbial entities differentially abundant in non-diabetic control: acetate (77.8%), $CO_2$ (55.6%), lactate (55.6%), propionate (44.4%) and butyrate (44.4%). (**d**) Top five most frequently produced metabolites by microbial entities differentially abundant in T2D: acetate (25.6%), $CO_2$ (21.4%), $NH_3$ (14.5%), $H_2S$ (14.5%) and ethanol (10.3%). Differences in these two sets of metabolites could suggest insights into metabolic processes associated with a T2D gut microbial ecosystem.

number of other microbial entities) prompted us to explore whether a hidden hierarchy of metabolic influence exists within the microbial communities. To this end, we introduce the T2D-relevant, community metabolic influence of a microbial entity $i$ ($\Phi_i$), which can be estimated as

$$\Phi_i \approx \sum_j H\left(\left|\left\langle \prod_{k,m \in P_{ij}} W_{km} \right\rangle_{P_{ij}} \frac{\Delta n_i}{n_i}\right| - \theta_\Phi\right),$$

where $H(\cdot)$ is the Heaviside step function, $W_{km}$ measures entity $k$'s direct metabolic influence on entity $m$ (as previously defined), $P_{ij}$ denotes one of the shortest paths connecting entities $i$ and $j$ in the influence network, $\langle\cdot\rangle_{P_{ij}}$ is the average over the shortest paths, $\Delta n_i/n_i$ characterizes entity $i$'s relative abundance change from control to T2D subjects, and $\theta_\Phi$ is a constant. For the full details of $\Phi_i$, including its formulation and case-dependent variations, see Methods and Supplementary Methods. Briefly, $\Phi_i$ denotes the cumulative number of community members towards which microbial entity $i$ exerts a very positive or negative metabolic influence in either a direct or indirect way (an indirect way here means exerting the influence through other intermediate microbial members; see Methods). For example, in our MIN, E. rectale was found to have a metabolic influence to 25 different microbial entities; hence, its community metabolic influence $\Phi_i$ is quantified as 25. Next, we measured the community metabolic influences of all microbial entities in MIN (Fig. 5a). In the MIN, we identified a set of highly interactive microbial entities, possibly acting as driving forces behind the global dynamics of their respective communities. We call these entities the network influencers. Among the total microbial entities, 17.6% were identified as network influencers.

Next, we examined some of the network influencers and the metabolites underlying their community metabolic influence towards other microbial entities (Supplementary Data 3). Influencers found to be abundant in T2D include Thermobifida fusca, A. muciniphila and Pseudomonas aeruginosa. T. fusca has a positive metabolic influence on non-influencers also abundant in T2D, by providing breakdown products of cellulose and hemicellulose. Likewise, A. muciniphila, through its participation in mucin degradation with other mucin degraders, has a positive metabolic influence to sulfate-reducing bacteria in MIN, by providing access to sulfate (a mucin degradation product). Qin et al.[12] (whose data we used in this study) also found an increased abundance of A. muciniphila in patients with T2D. However, Everard et al.[33,34] reported seemingly conflicting results, in which they observed a decrease in abundance of this mucin-degrader in faecal samples of diabetic mice. P. aeruginosa, a triglyceride degrader, can provide triglyceride degradation products to other microbial entities, indicating the role of dietary factors in this patient cohort. Influencers abundant in control (alternatively, scarce in T2D) include E. rectale and Streptococcus salivarius. E. rectale is a well-known producer of butyrate, which has been demonstrated to be capable of improving T2D-associated features[35–37]. S. salivarius was found to have overall negative metabolic influences toward many microbial entities scarce in control, with competition for the consumption of sugars and B vitamins.

To gain insight into more general properties of network influencers, we conducted a global analysis of associations between influencers and compounds. We found that the ability to degrade macromolecules (which in turn provides public goods) was the remarkably common metabolic feature among influencers. The average proportion of macromolecule degraders among influencers (82.6%) was ~10 times higher than that among non-

influencers (8.3%) (Fig. 5b), suggesting a prominent hallmark of network influencers.

Next, we identified microbes whose strains are recognized as human probiotics. Among the entities of this MIN, there were three probiotic species, all of which were non-influencers (L. delbrueckii, L. fermentum and Lactobacillus rhamnosus). Although the general tendency of this result has yet to be examined, this observation gives an interesting perspective on the future design of probiotic regimens. As an alternative to direct ingestion of multiple probiotic species, it may be advantageous to introduce influencers that could promote the growth of probiotic species already present among non-influencers. Therefore, a thorough understanding of metabolic influence among microbial community members could aid therapeutic methods aiming to modify gut ecological composition.

**Commonly produced metabolites by MIN microbial entities.** Metabolites that often originate from microbial entities abundant in T2D may be indicative of how gut microbes, through their metabolism, play key roles in a specific disease context. In this regard, we sought to identify the metabolites that are most commonly produced by microbial entities abundant in either T2D or control. Differences in these two sets of metabolites could provide insights into the metabolic processes of T2D-associated gut microbial ecosystems.

From all microbial entities differentially abundant in non-diabetic control, the top five commonly produced metabolites (in terms of the fraction of different entities that produce a given metabolite) were acetate (77.8%), $CO_2$ (55.6%), lactate (55.6%), propionate (44.4%) and butyrate (44.4%) (Fig. 5c). Butyrate and propionate have been shown to exert multiple beneficial effects on host physiology[38–41]. Some of these effects, which may contribute to protection from T2D, include intestinal glucose production (which helps prevent deregulation of glucose homeostasis and weight gain)[35], increase of energy expenditure[36] and anti-inflammatory effects[8,42]. In regards to lactate, it is noteworthy that certain strains of lactic acid bacteria have been reported to show anti-diabetic activities, possibly through a suppression of glucose absorption from the intestine[43].

On the other hand, the top five metabolites (in terms of the fraction of different entities that produce a given metabolite) produced by microbial entities differentially abundant in T2D were acetate (25.6%), $CO_2$ (21.4%), ammonia ($NH_3$) (14.5%), hydrogen sulfide ($H_2S$) (14.5%) and ethanol (10.3%) (Fig. 5d). Acetate and $CO_2$ overlap with those in the aforementioned case of non-diabetic control. For the remaining metabolites unique to T2D ($NH_3$, $H_2S$ and ethanol), their frequent appearance could suggest a distinct feature of the T2D-associated microbial community metabolism, although there is not yet conclusive evidence of their direct mechanistic links to T2D pathology. It still warrants mentioning that $NH_3$ and $H_2S$, often the outcomes of protein fermentation processes by intestinal bacteria, are known to cause adverse health effects as carcinogenic and genotoxic agents[44–47]. In the context of T2D pathology, it may be worthwhile to pursue these metabolites as part of future investigations into the mechanistic relationships between gut microbial metabolic processes and T2D.

**Discussion**
To provide a global framework for understanding community metabolism within the human gut, we have presented in this study NJS16, a network architecture encompassing the myriad relationships among gut microbial species, host cells and chemical compounds. Specifically, our network shows how individual microbes interact with their chemical environment

(via metabolite import, export and macromolecule degradation) and thereby with other microbes (via resource competition, interspecies cross-feeding and releasing macromolecule degradation products as public goods). One significant aspect of our work is the inclusion of microbe–host metabolic interactions. In this regard, our gut microbiota metabolite transport network can be investigated for identifying interaction pathways or modules that are associated with particular clinical conditions. As our network is mainly established upon literature annotations, it can serve as a useful data resource to those in the scientific community, who wish to gain insight into microbe–microbe and microbe–host metabolic interactions relevant to a particular context. Importantly, to stay scientifically correct and reliable, NJS16 will need to be routinely revised and augmented.

In a patient population with a specific set of socio-demographic characteristics (that is, male, mid-age and normal weight), the microbial entities abundant or scarce in T2D and the influence connections surrounding each microbial entity were shown in a community-scale MIN. The influence network suggests the presence of microbial entities that impose a relatively high degree of metabolic influence to other entities. These influencers are essentially the hubs of the network and could be considered as the main controllers of a hierarchical microbial community.

Apart from being consumed by microbial entities as part of an intricate cross-feeding web, metabolites that are produced and eventually exported into the microenvironment can directly affect host physiology through colonic absorption. $NH_3$ and $H_2S$, which are known to cause detrimental effects to host health[44–47], were found to be among the most frequently produced metabolites by microbial entities differentially abundant in T2D. Although further investigation falls outside the scope of this study, it would be intriguing to ask whether these metabolites might be contributing factors or molecular signatures of T2D conditions. However, focusing strictly on disease-associated metabolites may be limited in scope; ultimately, the origin of the exported metabolites (that is, the microbial entities) must be considered to fully explain the complex narrative of how the gut microbiota contributes to a disease state. To this end, our integrative framework can help elucidate metabolites that may be linked to a particular disease condition, their microbial sources and, importantly, the infrastructure of the entire community influencing the metabolism and growth of those microbial sources.

Several limitations of our study should be noted when interpreting our results. First, although the microbiome samples used in our study were metformin-naive and from a single ethnic background (Chinese), and our analysis was carefully conducted within a relatively homogeneous patient cohort, we cannot entirely exclude the possibility of other confounding factors. Notably, the gut microbiota can be significantly altered by one's dietary regimen[30,48], the information of which was not available in the original data set used in our study. Eventually, replicating our analyses on more finely classified patient cohorts—while maintaining sufficient sample sizes—could improve control for these potential confounders. In addition, in our study, a lack of time-series microbiome data for the individuals makes it hard to establish any clear causal relationship between their gut microbiota and disease. Availability of these time-series data would possibly lead to aetiological discoveries. Second, our gut microbiota metabolite transport network is currently limited to bacterial and archaeal species from a particular data source. It needs to be expanded towards other species from different data sources, as well as towards other major phylogenies such as eukaryotes and viruses. Recently, a taxonomic profiling method capable of strain-level identification of not only bacteria and archaea, but also eukaryotes and viruses, has been published[49].

Clearly, one can apply such newer methods to update and expand the coverage of microorganisms in our network. Third, almost all links between microbes and chemical compounds in our network were based upon literature annotations. We believe that this network gives us a higher quality data set than the ones obtained by purely bioinformatics predictions. However, our manual curation approach is not void of drawbacks: despite our best efforts, the manually curated network may involve possible misinterpretations of the literature information. In addition, many of the experimental evidences considered were from *in vitro* studies, of which the results may not be straightforwardly translated into the *in vivo* events inside the gut. Relatedly, oxygen-driven metabolic processes have long been thought to be irrelevant in the gut, but recent findings suggest the potential importance of oxygen for the gut microbiota composition[50–52]. NJS16 does not have oxygen as an explicit metabolic compound, although Supplementary Data 2 provides information on individual species' relationships with oxygen. Furthermore, the links between microbes and compounds in our network reflect simple binary information of either the presence or absence of the corresponding associations, whereby the degree of activity of those transport reactions, or individual organisms' growth requirements, are not yet distinguishable. Substrate-dependent product formation, interdependency of different metabolic pathways and end-product inhibition of cell growth have yet to be considered, and these would be better described by constraint-based genome-scale metabolic models. Taken together, our purely connectivity-based network structure should be considered as a map of the metabolic potential of the microbial community rather than of the actual state of metabolism itself. In addition, the abundance of literature annotations is clearly biased towards well-studied species and high-interest metabolites. These issues regarding the limited completeness of our network can be addressed to varying degrees, as a broader range and richer depth of literature evidence becomes available.

Although we acknowledge these limitations and challenges, we perceive them as guiding routes and benchmarks for improving our network. Each individual link in our network is from literature evidence (traceable literature references are provided in Supplementary Data 2), yet further experimental data are necessary to validate and update the global connectivity of the proposed network structure. Thus, our work calls for the need to develop high-throughput, quantitative techniques for identifying and validating specific functions (for example, import and export of metabolites and release of public goods from macromolecule degradation) and microbial metabolic interactions (for example, cross-feeding mechanisms and positive/negative metabolic influences) on a global scale, specifically within *in vivo* environments. Improvements in single-cell genomics and metabolomics strategies, in culturing techniques for previously uncultured microbes[53,54] and in platforms for *in vivo* high-throughput screenings will undoubtedly accelerate this process. The advent of such technologies could confirm or update our findings, as well as push forward and establish general concepts and theories of ecological systems biology.

Clearly, a comprehensive understanding of the metabolic relationships between gut microbes and of how those relationships are intertwined with host physiology is essential for the development of microbiota-based treatments for disease[55,56]. We see our work as an important step towards this direction, by providing an *in silico* platform for the rational design of microbial communities to benefit host health. Specifically, our network could be utilized to generate computational models for predicting the outcome of species-level perturbations on a microbial community and its host environment. Promising approaches in this front can be constraint-based methods[22,24,57–59] and kinetic

modelling[23,60–63]. In this direction, we expect our own microbial ecological networks to help in the development of personalized clinical strategies. Beyond metabolic interactions focused on by this study, considering quorum sensing molecules, virulence factors and genes for metabolic traits and transporters passed along in horizontal gene transfer[64,65] would be another interesting avenue to pursue using network-based approaches.

Ultimately, gut microbiome analyses will evolve beyond descriptive, profiling investigations towards more hypothesis-driven, mechanism-focused studies. However, progress in this direction will be contingent upon the maturation of our general understanding of the global inner workings of a microbial community. We envision that microbiome studies adopting systems biology approaches and multi-omics data integration[66] will be at the forefront of unraveling the complexity of the gut microbiota, as well as realizing its potential therapeutic applications.

## Methods

**Collection of microbiome data and taxonomic profiling.** Raw metagenomic sequencing data from faecal samples of 363 Chinese individuals (T2D and non-diabetic controls) were downloaded from the NCBI Sequence Read Archive (SRA) database (SRA045646 and SRA050230). Microbiome samples with missing patient metadata, of low read count after alignment (<100,000 reads), or highly deviated from the majority of the samples in their clustering results were removed from our study (Supplementary Data 1). The taxonomic profiling software that ran on these microbiome data was MetaPhlAn, which uses clade-specific marker sequences to identify microbial taxa (with species-level resolution) and their relative abundances from a metagenomic sample[67]. The marker gene catalogue used in MetaPhlAn was from microbial genomes from the Integrated Microbial Genomes system (July 2011). The MetaPhlAn software compares each metagenomic read from a sample to this marker catalogue, to identify high-confidence matches. When using MetaPhlAn, the default 'rel_ab' parameter options were selected. For all of our samples, using MetaPhlAn gave rise to a total of 1,219 identified bacterial and archaeal species, which cover ~70% of all the species from a unified gut microbiome data set with additional data sources (HMP reference genomes, HMSMCP–Shotgun MetaPHlAn Community Profiling and GutMeta DownLoad Center; accessed August 2016).

**Collection of metabolic information for NJS16 construction.** Metabolic information primarily used in this study was experimental evidence of metabolite transport or macromolecule degradation reported in literature. This information is dispersed across numerous scientific journal articles and textbooks. Therefore, a careful read of hundreds of these sources (Supplementary Data 2) was done to discern which annotations were experimentally verified, from those that were predicted solely based on automated bioinformatics algorithms. The small-molecule metabolites considered in our work were mostly primary metabolites, which are, nutrients involved in microbial growth, development or reproduction, or byproducts of those metabolic processes. Most chemical derivatives of those primary metabolites, as well as many secondary metabolites, for example, anti-microbial toxins, oligopeptides and quorum-sensing molecules, were excluded from our study. In addition, literature sources that report the messenger RNA or protein expression for metabolic-byproduct-producing enzymes, or for metabolite-specific transporters, were considered. Small metabolites that can diffuse through cell walls (for example, $H_2$ and $CO_2$), thereby not requiring transporter proteins, were also considered, as long as they serve as primary substrates and/or products of cellular metabolism. Furthermore, if a given microorganism or human cell type has a published, manually curated genome-scale metabolic model, transport reactions from the model were considered for that organism or cell type. Importantly, all annotated metabolite transport or macromolecule degradation processes for different strains of the same species were unified as collective features of that particular species. As degradation of a given macromolecule is generally conducted by multiple species in the gut, the corresponding degradation products (Supplementary Data 2) were considered as indirect export products of all species involved in that macromolecule degradation.

In NJS16, one set of nodes corresponds to organisms in the gut microbiota community (that is, microbial species and host cells), whereas the other set corresponds to chemical compounds (small-molecule metabolites or macromolecules). The microbes in our network were connected to the metabolites they can import from, and/or export to, their microenvironment, or to the macromolecules they can degrade in their extracellular space. Taken together, NJS16 is a comprehensive, primarily literature-curated, microbiota interaction map that accounts for 567 bacterial and archaeal species in the large intestine, 3 human cell types metabolically interacting with those colonic microbes and 244 chemical compounds—all interconnected via 4,483 small-molecule metabolite transport or macromolecule degradation processes. To facilitate visual exploration,

Supplementary Data 4 provides NJS16 in graph-editor accessible, markup language file format.

**Identification of patient-cohort-specific microbial entities.** All microbiome samples were categorized into sub-population cohorts based on each subject's gender (male/female), age range (young (age < 45 years), mid-age (45 years ≤ age < 65 years) and old age (65 years ≤ age), as defined by the United States Census Bureau), and body mass index (BMI) range (underweight (BMI < 18.5), normal weight (18.5 ≤ BMI < 25) and overweight (25 ≤ BMI)). Among our T2D samples, we used only those from metformin-untreated patients, information on which was provided by a recent study on metformin-confounding effects on the gut microbiota[32]. Among all 18 possible cohorts for both T2D patients and non-diabetic controls, the male, mid-age and normal weight cohort (control $n = 21$; T2D $n = 11$) was selected for detailed analyses based on its having the largest sample size among all cohorts, as well as having comparable sample numbers in both phenotypes (control and T2D). Microbial species differentially abundant or scarce in T2D patients (compared with non-diabetic control subjects) were identified by the Wilcoxon rank-sum test and false discovery rate (FDR) correction was used for multiple testing (FDR < 0.1; Supplementary Methods). To identify differentially abundant or scarce groups (genus or metabolic clique), the sum of all relative abundances of microbial species in a particular group was obtained to represent the relative abundance of the group, then followed by the Wilcoxon rank-sum test and false discovery rate correction (FDR < 0.1; Supplementary Methods). The microbial entities (species, genera and metabolic cliques) found to be differentially abundant or scarce in T2D were selected for further pruning to identify the most representative microbial entities (Supplementary Methods).

**Construction of a microbial MIN.** The bipartite network of microbial species and metabolic compounds (NJS16) can be utilized as a global template for constructing a context-specific network. For the male, mid-age and normal-weight cohort, NJS16 was converted into a unipartite network of microbial entities (species, genera and metabolic cliques) differentially abundant or scarce in T2D compared with non-diabetic control. This network is called herein a microbial MIN. The conceptual framework for constructing this network is as follows: in complex, microbial ecosystems, a microbial entity can provide nutrients to another entity via interspecies cross-feeding of metabolic byproducts and/or release of macromolecule degradation products. This positive impact may potentially promote microbial growth. In contrast, a microbial entity can limit the access to nutrients of another entity via competition for the same metabolites. This negative impact may potentially inhibit microbial growth. Based on the combination of these positive and negative effects, the net metabolic influence of a microbial entity $i$ on another entity $j$ ($W_{ij}$) can be calculated. To quantify $W_{ij}$, the potential metabolic influence of one microbial species on another microbial species needs to be estimated. For a pair of species $i$ and $j$, an increase in species $i$'s abundance may contribute to an increase or decrease in species $j$'s growth (the use of either the growth rate or the abundance does not much affect the following argument). Species $i$'s influence on species $j$'s growth is expressed by $W_{ij} = \frac{(\partial \mu_j)/\mu_j}{(\partial n_i)/n_i}$, where $\mu_j$ is species $j$'s growth rate and $n_i$ is species $i$'s abundance. $W_{ij} > 0$ (< 0) indicates the positive (negative) metabolic influence that species $i$ is promotive (inhibitive) for species $j$'s growth. In more detail, $W_{ij} = \frac{(\partial \mu_j)/\mu_j}{(\partial n_i)/n_i} = \frac{n_i}{\mu_j} \cdot \left( \frac{\partial \mu_j}{\partial n_i} \right) = \frac{n_i}{\mu_j} \cdot \sum_k \frac{\partial \mu_j}{\partial z_{kj}} \frac{\partial z_{kj}}{\partial n_i}$, where $k$ denotes each metabolite imported by species $j$ and $z_{kj}$ is the rate of metabolite $k$ consumed by species $j$ per unit abundance. By assuming proper forms of $\mu_j$ and $z_{kj}$, as a function of $\{z_{kj}\}$ and that of $\{n_i\}$, respectively (Supplementary Methods), $W_{ij}$ can be written as:

$$W_{ij} \approx \sum_k^{g_{ik}^c = 1} \left\{ \alpha \frac{n_i g_{ik}^p}{\sum_m n_m g_{mk}^p} - \frac{n_i g_{ik}^c}{\sum_m n_m g_{mk}^c} \right\},$$

where $g_{ik}^{p(c)} = 1$ if species $i$ produces (consumes) metabolite $k$, or otherwise $g_{ik}^{p(c)} = 0$ and $\alpha$ is a constant. If species $i$ degrades macromolecules whose breakdown products are consumed by species $j$, $W_{ij}$ contains additional terms that are described in Supplementary Methods. Further details of $W_{ij}$, including its formulation, the incorporation of macromolecule degradation and the determination of $\alpha$, are presented in Supplementary Methods.

Next, we found that a potential metabolic influence of one microbial group $G$ on another microbial group $\Gamma$ (each group is either a genus or metabolic clique) is the weighted sum of $W_{lq}$ (species $l$ in group $G$ and species $q$ in group $\Gamma$) with each weight being proportional to species $q$'s abundance (Supplementary Methods). Consequently, a metabolic influence of one microbial entity on another microbial entity can be calculated for every pair-wise case. For every pair of differentially abundant or scarce entities $i$ and $j$ in T2D, a directed link with weight $W_{ij}$ from entity $i$ to entity $j$ (and vice versa) was assigned. Among these links, only the links that account for actual microbial abundance changes between control and T2D subjects (Supplementary Methods) were considered.

In addition to microbe–microbe interactions, microbe–host interactions were examined and the representative interactions in the patient cohort were identified as follows: among metabolic cliques that are differentially abundant in T2D or in control, those that could either directly influence, or are directly influenced by, host

cells were initially selected. These metabolic cliques can be regarded as proxies for enriched chemical compounds in T2D or in control. Then, all microbial entities (of the corresponding MIN) with commonalties to the previously selected cliques in regards to relevant phenotypes (abundant in T2D or in control), as well as metabolic capability (consumption, production or degradation of a chemical compound), were selected. These microbial entities were recognized as putative candidates for having an important metabolic relationship with the host. Full details of our methods are presented in Supplementary Methods.

**Identification of network influencers.** In a microbial MIN, a microbial entity $i$ can exert a metabolic influence on another microbial entity $j$ if they are directly connected to each other. Even when they do not have a direct connection, entity $i$ may indirectly exert a metabolic influence on entity $j$, if other entities located between the two can transfer entity $i$'s influence to entity $j$. Taking into account both of these direct and indirect effects, the quantity $\Phi_{ij}^{P_{ij}}$, which measures entity $i$'s influence on entity $j$ along the shortest path $P_{ij}$ between the two entities, can be defined (we only consider the influences that are relevant to the host phenotype difference between T2D and control). As there can be either single or multiple shortest paths, $\Phi_{ij} \equiv \langle \Phi_{ij}^{P_{ij}} \rangle_{P_{ij}}$, where $\langle \cdot \rangle_{P_{ij}}$ is the average over the shortest paths. $\Phi_{ij}$ serves as an estimate of entity $i$'s overall metabolic influence on entity $j$.

Next, the community-level metabolic influence of entity $i$ ($\Phi_i$) is defined as the number of entity $j$'s that satisfy $|\Phi_i| \geq \theta_\Phi$ ($\theta_\Phi$ is a constant determined in Supplementary Methods). By formulating $\Phi_{ij}^{P_{ij}}$ as a function of $\{W_{km}: k, m \in P_{ij}\}$ and microbial abundances ($W_{km}$ denotes entity $k$'s direct metabolic influence on entity $m$, as previously described), $\Phi_i$ can be written as:

$$\Phi_i \approx \sum_j H\left( \left| \left\langle \prod_{k,m \in P_{ij}} W_{km} \right\rangle_{P_{ij}} \frac{\Delta n_i}{n_i} \right| - \theta_\Phi \right),$$

where $H(\cdot)$ is the Heaviside step function and $\Delta n_i/n_i$ characterizes entity $i$'s relative abundance change from control to T2D subjects. For full details of $\Phi_i$, including its formulation and case-dependent variation, see Supplementary Methods.

Lastly, the network influencers, which are microbial entities with distinctively large $\Phi_i$'s, were identified as follows: we identified a transition point of $\Phi_i$ from the probability distribution of $\Phi_i$ (Fig. 5a), which distinguishes one group of microbial entities (shaded area in Fig. 5a) from the other in their $\Phi_i$'s, and we use this transition point of $\Phi_i$ as the lower bound of the influencers' $\Phi_i$.

**Data availability.** NJS16 (Supplementary Data 2) is also available in the Dryad Digital Repository[68]. Codes for the computation of microbe–microbe and microbe–host metabolic influences are available as Supplementary Software. The authors declare that all other relevant data are available within the article and its Supplementary Information files, or from the corresponding author upon request.

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

## Acknowledgements

We thank H. Bjørn Nielsen, Matthew Benedict, Matthew Gonnerman and Caroline B. M. Porter for useful discussions. This work was supported by the National Research Foundation of Korea (NRF) Grant NRF-2015R1C1A1A02037045 (J.S., S.K., J.J.T.C. and P.-J.K.) and NRF-2015R1A2A1A10056126 (S.J. and G.Y.J.), and the Advanced Biomass R&D Center (ABC) of Global Frontier Project ABC-2015M3A6A2066119 (S.J. and G.Y.J.) funded by the Korean Government (MSIP). This work was also supported by the ICTP through the OEA-AC-98 (S.K.), the Mayo Clinic Center for Individualized Medicine (N.C.) and the National Institutes of Health under award number R01CA179243 (N.C.).

## Author contributions

J.S. and P.-J.K. designed the research. J.S., S.K., J.J.T.C. and S.J. performed the research. J.S., S.K., J.J.T.C., S.J., Y.-S.J., G.Y.J., N.C. and P.-J.K. analysed the data. J.S., S.K., N.C. and P.-J.K. wrote the manuscript.

## Additional information

**Competing interests:** The authors declare no competing financial interests.

