## [Peer Review File · Nature Communications]

Reviewers' comments:

Reviewer #1 (Remarks to the Author):

The manuscript by Sung et al. addresses an important and timely question of metabolic interactions in gut microbiota. To this end, they reconstruct a literature-guided network built upon metabolite-bacteria relations. Although the network has value as a resource, the claimed biological value is overestimated and many important confounders and aspects are neglected.

1. The T2DM microbiota is known to be heavily influenced by metformin. A recent study (Forslund et al., 2015, Nature) shows that the T2DM signatures in these datasets (including the one the authors in this study use) are basically metformin signatures.

2. Although the literature-based network is a commendable effort, it ignores the many other metabolic capabilities of microbes which are better captured in conjunction with genome-scale metabolic models. The claim by the authors that their network model is more "mechanistic" is puzzling as the literature data on consumed or produced metabolites is often in vitro, context dependent and represents only a small fraction of the metabolic capabilities of any microorganism. The network approach also ignores the interdependencies between different metabolic pathways and growth requirements of individual species.

3. The overall conclusion is speculative (pertaining to the descriptive nature of the results, which is basically the reconstructed network). What are the specific insights or principles revealed by the study? Why the analysis is restricted to only T2DM subjects?

Reviewer #2 (Remarks to the Author):

The paper discusses the creation of a high-quality network of the human gut microbiota (NJS16), with ~600 microbial species and 3 human cell types metabolically interacting through >5,000 small-molecule transport and macromolecule degradation events. Using this network, the authors investigated the disease-related infrastructure of the gut microbial ecosystem in type 2 diabetes (T2D) patients.

In general, such a repository can assist in deducing the interactions between gut biota members in health and disease states, and can also be a reference point when manually or automatically creating metabolic models for the ~600 microbial species members of the gut biota.

Major remarks:

1. The first question to ask is "what is the importance of the NJS16 compilation besides being a reference point". For the last few years much of the research done on the metabolic aspects of microbial species in general including their interactions within the gut flora has been done using constraint based metabolic models. In these models the "Exchange metabolites" (both those up-taken or secreted) are listed. It is clear from reading the bibliography of the paper, that the authors are aware of this. In supp Figure 1, the authors partly address this issue by showing that the deduced transportable metabolites lists from "KEGG" for all gut species is much bigger and included the ones listed in NJS16 for the same species. The authors state that there might be false positives in KEGG (i.e., metabolites that are not transportable, and appear in the lists of transportable metabolites for specific organisms using KEGG), however they do not give a clue to how to identify these metabolites. There might also be (and probably there are) also false negatives in the NJS16 (i.e., metabolites that should be in the transportable lists for specific

organisms, and do not appear in NJS16). The authors do not help identifying those metabolites too. It is not clear if the missing metabolites in NJS16 do not cause more damage in predicting the gut flora behavior in health and disease states, compared to the wrongfully added metabolites in the KEGG deduced matrix. An analysis to clarify the predictability accuracy issue should be done (see point 6).

I suggest repeating the analysis done in Sup Figure 1 for KEGG, on metabolic models created by the "KBASE" platform, and for matrices deduced from other GPR databases.

Another analysis that could be done to learn more about the coverage of up-taken metabolites by the species is to compare the up-taken metabolites with known growth media.

2. It is clear that metabolites secreted from given microorganisms are tightly connected to the nutritional resources available at the gut, at least to some extent; there is no reference in the repository to the diet, and to whether the secreted metabolites are related to up-taken metabolites for specific microbes.

3. In order for this resource to be useful, a reference to known databases IDs should be given both for the organisms (using the NCBI taxonomy tree for example) and for the metabolites (using the KBASE or KEGG repositories). Without this, the resource utility will be quite limited.

4. There is a claim that small molecules (small metabolites) can diffuse through the cell-walls, this point is not addressed in the paper.

As for the conclusions drawn from NJS16 with regard to the T2D:

5. The selected groups of patients and controls are missing important information needed to reach conclusions regarding metabolic behavior of gut flora (the authors mention that too in the body of the article):

a. What is the diet of the people in the group? This has a major impact on the structure of the gut flora. I suspect that the diet has much bigger affect on the gut flora, than the one that T2D has.

b. What is the medical treatment that the people in the groups have received? It is known that antibiotics and anti diabetic treatments affect the gut flora (as the authors state).

These points should be validated using T2D and control groups with controlled diet & medication.

6. As there is already published work on cooperation and competition between species communities done on the level of genome scale metabolic models (GSM) (as can be found in the authors bibliography), a comparison of the results deduced from NJS16 and the ones from the GSMs should be done. A comparison of the results deduced should also be done between NJS16 and the matrices deduced from other GPR databases (KEGG, etc). It is essential to see which have more predictability.

Reviewer #3 (Remarks to the Author):

In this study, the authors have generated a theoretical human gut reference metabolic network from literature and provide this as a resource for the systems biology community. They also applied this network and newly developed bioinformatics methods to re-examine the Qin et al Chinese diabetes study.

Having a complete, reliable, integrated metabolic network describing all bacteria-bacteria and bacteria-host interactions in the human gut is essential for developing systems-based approaches to analyse and predict gut microbiota behavior, just as the first reference models were for bacterial

or human systems biology. So, if done properly, such a network is an important resource to the field. Given the impressive amount of work (literature study of several hundreds of articles) I was very enthusiastic that this study would be describing that resource. However, a more in depth analysis and some 'sample checks' indicate to me that the network, as it stands, is still far from what it needs to be. I could only highlight some examples, but if my quick checks already raise that many eyebrows, I gravely worry about the quality of the rest of the network. Given the fact that many groups with little knowledge on the gut microbial processes will use this resource as is, I currently have no option than to strongly advice against publishing this paper and resource, until it has been properly vetted and validated. Below, I give a number of criticisms and suggestions towards that aim.

- The starting point of this study is the metagenomic data of the chinese diabetes cohort (qin et al); if the goal of the authors is to provide a general-purpose tool, also the data generated by the HMP and MetaHIT efforts (as well as those of e.g. the appropriate Gordon and Knight lab studies and the recent large-scale population studies) should be used. The species list generated from these data would then be truly representative for the gut diversity and be a sound basis for the network which would be applicable across continents.
- The reference list contains papers on animal microbiotas, food fermentation etc - it might be that some of these environments contain similar species as those present in the human gut, but its far from certain that the human strains have or perform the same functionality as those from those papers. So despite the obviously impressive amount of work, I worry that the current network is ridden with reactions for which it is uncertain they actually happen leading to noisy and potentially biologically irrelevant results. I suggest the authors focus on human-only reactions to avoid this.
- Similarly, what about oxygen? Some of the bacteria/processes described in the tables are strictly aerobic (acetate production by acetic acid bacteria such as acetobacter). The gut ecosystem is mostly anaerobic so these processes are very unlikely to be important if present at all.
- The authors assign functional capacities from papers (at species level I assume) to metagenome-derived phylogenetic annotations. Strain level variation and environmental context can be a in important cause of uncertainty here. The authors do not perform any validation of their whole procedure, in silico nor in vitro. For none of the reactions in the network we have an assessment of its reliability, its potential to actually happen in the gut etc - it all comes down to a very greedy, noise-sensitive approach, which is likely to result in a network ridden with errors.
- To give just one more example: What is the importance of some of the processes described in the tables for the gut? Ammonia oxidation? Denitrification? Nitrogen fixation? Are they present at all?
- Another one: the authors describe 60 archaea in their list, which is highly unlikely - they are probably found because of mismapping in the taxonomic annotation. I suggest the authors revisit the species mappings of the metagenomic datasets they start from and be extremely conservative (ie more than the original authors); they are more likely to benefit from a smaller, yet more high-quality network than a larger but noisier one.
- Also the quality of the literature study itself worries me: for instance, *B. longum* is mentioned as a hydrogen producer, based on Barcinella et al. 2000. In that paper, *B. longum* is also reported to produce butyrate. Both results are unlikely, but hydrogen was included in the database, butyrate not (although both are obviously related). Why?
- Import/export of metabolites is notoriously difficult to determine as transporter specificity can be extremely substrate specific. The authors seem to equate breakdown/production with import/export, yet this is far from always the case. Can the authors provide more detail of the type of evidence they used to determine exchange of metabolites between species?
- Likewise, there is something called end-product inhibition. Two bacteria producing the same end-product limit each others growth. A classic example is hydrogen (but not in all cases) - but this is not taken into account in the network. Also, there is a link between substrate fermentation and nature and amount of metabolites produced. By splitting into import and explort, this link was cut by the researchers, which I don't think is a good representation of the truth.
- With regard to the metformin confounder, did you contact the original authors of the diabetes study to obtain metformin usage data? The diabetes results are unlikely to be very robust without

that knowledge.

- P7-8: Diabetes results - I actually wonder why one needs a metabolic network to find the results described here. Can't these results be found by simple differential pathway analysis?

- I'm not sure how to interpret the data in dataset 3 B. What with conflicting signs columns F and G? See also discussion on page 7. E. rectale is reported to compete with M. paludicola for hydrogen gas, but, to my knowledge, rectale produces hydrogen. Also, M paludicola does not seem to appear in dataset 2 B. Why?

- I would avoid classifying metabolites as 'toxic'. That depends entirely on context and concentration.

- In supplemental Table 2, sheet b, more background on the literature-reported relationships would be helpful to know how reliable the experiment is. For instance, the sheet could at least report the experimental system (animal model, bioreactor, plate, etc.) and in future could report more details, such as medium, pH, temperature, atmosphere, organism grown alone or in a mixture etc., all of which may affect which metabolites are produced and consumed.

- W_{ij} does not weight the importance of metabolites for the survival of species j. The metabolic influence of species i on j will be a different one depending on whether or not a metabolite produced by i can be replaced by another metabolite in i's absence. Also, species may produce and consume different metabolites depending on the concentration of these metabolites, i.e. they may exhibit different metabolic strategies. The predictions of influencers have therefore to be experimentally validated and should not be presented as final.

- The main network is summarized in supplemental Table 2, sheet f. As a courtesy to readers that want to explore this network, please provide it in a format easily readable by graph editors (e.g. xgml), with at least one node attribute that provides the node type (metabolite or microorganism) and the consumption/production status and, if possible, the literature source as edge attributes. Additional node attributes could include the lineage for microorganisms and the KEGG identifier for metabolites.

- On page 6/17, m is not explained. Is m referring to all species that can produce/consume metabolite k?

- I'm not sure how to interpret the data in dataset 3 B. What with conflicting signs columns F and G? See also discussion on page 7. E. rectale is reported to compete with M. paludicola for hydrogen gas, but, to my knowledge, rectale produces hydrogen. Also, M paludicola does not seem to appear in dataset 2 B. Why?

- "Each species in NJS16 imports 5.2 and exports 3.7 metabolites" - the median is a more meaningful average than the mean here

- Overall the paper could be better written, especially given the broad scope that NCOMMS aims at. Its very dense & complex with a lot of different messages in many directions - overall it could benefit from a major overhaul to increase accessibility for a wider audience.

Response to Reviewer #1

We are very thankful for the careful suggestions and comments offered by the Reviewer. The Reviewer appreciated our study as “*The manuscript by Sung et al. addresses an important and timely question of metabolic interactions in gut microbiota.*”

We carefully considered the Reviewer’s suggestions for the revision of our manuscript. In the following, we discuss the Reviewer’s comments and the changes we made in the manuscript.

“1. *The T2DM microbiota is known to be heavily influenced by metformin. A recent study (Forslund et al., 2015, Nature) shows that the T2DM signatures in these datasets (including the one the authors in this study use) are basically metformin signatures..*”

We appreciate the Reviewer’s excellent comment. As the Reviewer pointed out, during the preparation of our original manuscript, Forslund *et al.* published a work about metformin signatures in T2D microbiome samples (we also noted this in our original manuscript). At the time when we originally analyzed the dataset from Qin *et al.*’s study (2012), information on whether a given microbiome sample was from a metformin-naïve or metformin-treated patient was not publicly available. Now, following the Reviewer’s excellent comment, we have looked into Forslund *et al.*’s study, and found that metformin-naïve or metformin-treated information for patient microbiome samples in Qin *et al.*’s study is currently provided. Based on this information, we have *re-performed our entire analysis* with the Qin *et al.*’s metformin-naïve T2D samples. In this analysis, metformin-treated samples were excluded in order to avoid any metformin-confounding effects that the Reviewer pointed out. Among all possible cohorts for both metformin-naïve T2D patients and non-diabetic controls, we selected four cohorts of sufficient and similar sample sizes for parallel analyses (in the original manuscript, we had selected six cohorts, but we have now selected four cohorts because of this criteria of having sufficient sample sizes from metformin-naïve patients). These cohorts are (i) male, young, normal weight (Cohort 1); (ii) male, mid-age, normal weight (Cohort 2); (iii) female, mid-age, normal weight (Cohort 3); and (iv) female, mid-age, overweight (Cohort 4). Full details of the cohort categories are presented in Methods and Supplementary Data 1 of our revised manuscript. With these metformin-naïve samples, our new analysis shows that the main conclusions and insights of our manuscript do not change much from the previous ones, despite some differences in the details of the results. For example, one can compare the following Fig. 3 of the revised manuscript with Fig. 3 of the original manuscript, and can still find remarkable similarities.

Thanks to the Reviewer's above valuable comment, we have thoroughly adopted the new, metformin-free, analysis results in our revised manuscript.

"2. Although the literature-based network is a commendable effort, it ignores the many other metabolic capabilities of microbes which are better captured in conjunction with genome-scale metabolic models. The claim by the authors that their network model is more "mechanistic" is puzzling as the literature data on consumed or produced metabolites is often in vitro, context dependent and represents only a small fraction of the metabolic capabilities of any microorganism. The network approach also ignores the interdependencies between different metabolic pathways and growth requirements of individual species."

We thank the Reviewer for the above valuable points. We fully agree that our primarily literature-based network might account for only a part of the whole metabolic capability of each microorganism, and has notable limitations in its true mechanistic power. Indeed, a more promising approach can be the use of genome-scale metabolic models that the Reviewer mentioned. In fact, as described in Methods (page 15) of our original manuscript, if

a given microorganism or human cell type has a published, manually-curated genome-scale metabolic model, we incorporated transport reactions from the model into our network, NJS16. Nevertheless, except for ~40 microbial species whose manually-curated genome-scale metabolic models were publicly available during our NJS16 construction stages, the vast majority of microbial species in our network do not have such models readily available. Given this current research community's situation that only a tiny fraction of human gut microbes has manually-curated genome-scale metabolic models, we deemed that our next best strategy was to harness experimental evidence-derived literature information for the remaining species in our network. Alternatively, one may ask us to consider using automated reconstruction of genome-scale metabolic models (like those provided by KBase), as a way to overcome the aforementioned scarcity of manually-curated genome-scale models; however, we did not use such automated strategies because of their highly error-prone identification of transportable metabolites. In sum, despite several limitations, we determined that a primarily literature-based network of the human gut microbiota was our best practical choice to pursue in our study. Therefore, a large body of our network can then be viewed as an integration of current biological or experimental knowledge. Because of its globally-organized network nature, one can still apply an array of powerful network structural analysis methods developed in the network science and pathway analysis communities [e.g., *Nature Reviews Genetics* **12**, 56 (2011); *Science* **347**, 1257601 (2015)].

In line with the Reviewer's excellent comments, we would like to see our network serves as a valuable resource to aid the generation of future genome-scale metabolic models. In the Supplementary Data 2 of our revised manuscript, we have added the KEGG compound ID of each transportable metabolite as a service to researchers who use this database when building genome-scale metabolic models. In addition, as the Reviewer pointed out, we have revised our manuscript to minimize the use of the word "mechanistic" when describing our network. As a result, this word is used only at absolutely necessary places in our revised manuscript.

In regards to the rest of the Reviewer's above comments, we have discussed them in our revised manuscript, as follows:

(Revised, pages 12–13) "However, our manual curation approach is not void of drawbacks: ... many of the experimental evidences considered were from *in vitro* studies, of which the results may not be straightforwardly translated into the *in vivo* events inside the gut. Furthermore, the links between microbes and compounds in our network reflect simple binary information of either the *presence* or *absence* of the corresponding associations, whereby the degree of activity of those transport reactions, or individual organisms' growth requirements, are not yet distinguishable. Substrate-dependent product formation, interdependency of different metabolic pathways, and end-product inhibition of cell growth have yet to be considered, and these would be better described by constraint-based genome-scale metabolic models. Our purely

connectivity-based network structure should be considered as a map of the *metabolic potential* of the microbial community, rather than of the actual state of metabolism itself. ... Each individual link in our network is from literature evidence ... yet further experimental data may be necessary ... Improvements ... in platforms for *in vivo* high-throughput screenings will undoubtedly accelerate this process.”

“3. *The overall conclusion is speculative (pertaining to the descriptive nature of the results, which is basically the reconstructed network). What are the specific insights or principles revealed by the study? Why the analysis is restricted to only T2DM subjects?*”

These are indeed important questions. Among the many results in our study (e.g., network influencers, cohort-specific network features, and commonalities in T2D-associated metabolic functions), we thought that the following points are the most intriguing. Our community-level network framework for gut microbiome analysis revealed noteworthy connections among interspecies cross-feeding pathways in T2D; nontoxic, rather than toxic, metabolites are likely to maintain the structural integrity of T2D-specific microbial communities (Fig. 3e), and thus could be implicated in the pathology of this disease. Therefore, **metabolites seemingly nontoxic or even beneficial to the host could be detrimental to host health** if their community-level effects are considered (pages 9–10). These metabolites include **B vitamins, such as pyridoxine, riboflavin, and folate**, from our results (page 10). In this sense, we suggest a counterintuitive possibility that **some microbes, which were originally conceived to promote health by producing vital nutrients, may actually be nurturing a microbial community associated with disease.**

Accordingly, we have improved our manuscript by more clearly stating these points, as follows:

(Revised, abstract) “A system-level framework of complex microbe-microbe and host-microbe chemical cross-talk would help elucidate the role of our gut microbiota in health and disease ... Intriguingly, in addition to secreting toxic metabolic products, microbes may contribute towards disease with their nontoxic or even host-beneficial products (such as B vitamins), by nurturing interspecies cross-feeding pathways that maintain T2D-specific gut ecosystems.”

(Revised, page 3) “Combined with fecal metagenomic information from T2D patients¹², our network analysis reveals an intriguing metabolic infrastructure of the T2D gut ecosystem. Remarkably, our results suggest that microbial products seemingly nontoxic or even beneficial to the host, such as B vitamins, may still contribute towards disease pathology through their community-level roles in the

maintenance of a T2D-specific gut ecosystem.”

(Revised, page 10) “Hence, some metabolites seemingly nontoxic or even beneficial to the host could be detrimental to host health if their community-level consequences are considered. Interestingly, we found that these metabolites include B vitamins, such as pyridoxine, riboflavin, and folate (Table 1 and Supplementary Table 1). In this sense, our results suggest a counterintuitive possibility that some microbes, which were originally conceived to promote health by producing vital nutrients, may actually be, in some cases, nurturing a microbial community conducive to disease.”

Other parts of our manuscript have also been revised to enhance overall clarity. In addition, we have made the contents of the manuscript more concise. For example, mathematical formulas in pages 6 and 8 of the original manuscript are now entirely covered by Methods and Supplementary Methods in our revised manuscript.

The Reviewer asked why our analysis is restricted to T2D microbiome samples. Surely, our method can be applied to a broad range of gut microbiota and host conditions. In our manuscript, we focused on T2D, not only as one such application, but also to help address a critical need. T2D is the most prevalent endocrine disease and represents one of the most important global health challenges of this century; an estimated total of 387 million people in the world (9% of the adult population) have diabetes, of which 90% corresponds to T2D. Moving forward, we expect that many follow-up studies will adopt our community-level-focused approach to investigate microbiomes associated with other diseases.

The additional changes made in the revised manuscript are the following:

1. We have added the following statement to clarify our network construction method:

(Revised, page 14) “Small metabolites that can diffuse through cell walls (e.g., H₂ and CO₂), thereby not requiring transporter proteins, were also considered, as long as they serve as primary substrates and/or products of cellular metabolism.”

2. We have added the following analysis result for our network:

(Revised, page 4) “[interestingly, our network and microbiome data show that the similarity in two species’ nutritional profiles is positively correlated with the species’ co-occurrence ($\rho = 0.23$ and $P = 0.03$), in agreement with a previous claim²⁰ when the same measures were applied].”

3. We have revised our network to improve its quality, mainly by correcting for ambiguous literature information. All the results presented in our revised manuscript have been generated by using this improved version of our network. Supplementary Data 2 provides the details of our improved network, along with a traceable reference list.
4. In Supplementary Fig. 1b, we have analyzed known growth media information. In addition, in Supplementary Data 2, we have separately marked food macromolecules, vitamins, and other compounds that can be derived from host diet.
5. We have examined the coverage of a species list from Qin *et al.*'s data, as follows:

(Revised, page 14) “For all of our samples, using MetaPhlAn gave rise to a total of 1,219 identified bacterial and archaeal species, which cover ~70% of all the species from unified gut microbiome data with additional data sources (HMP reference genomes, HMSMCP - Shotgun MetaPHlAn Community Profiling, and GutMeta DownLoad Center; accessed August 2016).”

6. In Supplementary Data 2, we have separately marked an extremely “conservative” species list. This list consists of microbial species that have been previously well studied regarding their relationship to the human gut. Supplementary Fig. 3 and Supplementary Table 1 show that our main analysis results do not change much qualitatively between the networks with our full and conservative species lists.

Response to Reviewer #2

We are very thankful for the careful suggestions and comments offered by the Reviewer. The Reviewer appreciated our study as *“In general, such a repository can assist in deducing the interactions between gut biota members in health and disease states ...”*

We have carefully examined the Reviewer’s comments, and have revised our manuscript accordingly. Discussions on their comments and on the relevant changes we made to our manuscript are provided below.

“1. The first question to ask is “what is the importance of the NJS16 compilation besides being a reference point”. For the last few years much of the research done on the metabolic aspects of microbial species in general including their interactions within the gut flora has been done using constraint based metabolic models. In these models the “Exchange metabolites” (both those up-taken or secreted) are listed. It is clear from reading the bibliography of the paper, that the authors are aware of this. In supp Figure 1, the authors partly address this issue by showing that the deduced transportable metabolites lists from “KEGG” for all gut species is much bigger and included the ones listed in NJS16 for the same species. The authors state that there might be false positives in KEGG (i.e., metabolites that are not transportable, and appear in the lists of transportable metabolites for specific organisms using KEGG), however they do not give a clue to how to identify these metabolites. There might also be (and probably there are) also false negatives in the NJS16 (i.e., metabolites that should be in the transportable lists for specific organisms, and do not appear in NJS16). The authors do not help identifying those metabolites too. It is not clear if the missing metabolites in NJS16 do not cause more damage in predicting the gut flora behavior in health and disease states, compared to the wrongfully added metabolites in the KEGG deduced matrix. Analysis to clarify the predictability accuracy issue should be done (see point 6).

I suggest repeating the analysis done in Sup Figure 1 for KEGG, on metabolic models created by the “KBASE” platform, and for matrices deduced from other GPR databases.

Another analysis that could be done to learn more about the coverage of up-taken metabolites by the species is to compare the up-taken metabolites with known growth media.”

These are very excellent comments. It should be noted that, when we started our network construction, we initially did consider the use of bioinformatics-based prediction methods to infer transportable metabolites, as in the Reviewer’s reasonable suggestions. However, we swiftly noticed how error-prone those results were. Compared to annotations of metabolic

enzymes that have been of intensive interest, annotations of transporters are still too far from complete. To see how unspecific and ambiguous current substrate annotations of transporters are, one can readily check TransportDB. Furthermore, there are frequent and obvious defects in the existing bioinformatics-based methods to infer transportable metabolites. Using KEGG annotations, we tested one such method [*PNAS* **110**, 12804 (2013)], on the microbial species in our network. Based on our calculations, we found that glucose 6-phosphate, fructose 6-phosphate, and fructose 1,6-bisphosphate were suggested to be some of exportable compounds by this method. Besides this very unrealistic result, we found that the outcomes varied too much across species, even in those belonging to the same genus, questioning their overall biological relevance. Not only false positives, but also the impact of false negatives can be problematic with bioinformatics-based approaches: following the Reviewer's careful comment, we tried to use KBase to identify transportable metabolites for our microbial species. We found that KBase does not identify fundamental fermentation products of numerous microbial species, such as acetate, lactate, formate, and ethanol for *Bifidobacterium breve*. Supposedly, these prevalent failures to capture biological knowledge could be the tip of the iceberg, indicating fundamentally serious false positive and false negative errors inherent to current bioinformatics-based methods. Therefore, we conclude that any direct comparison to these bioinformatics-based results would not be suitable to determine the accuracy of our network, NJS16.

Our network is mainly established upon literature annotations, and can hence be viewed as a collection of current biological knowledge or curated experimental data. We completely agree with the Reviewer that our mainly literature-based network might account for only a part of the whole metabolic capability of each microorganism. Nevertheless, as new experimental data and biological knowledge become available, we expect that NJS16 will continue to be updated into a more comprehensive network. Also, we agree to the Reviewer's opinion on the utility of constraint-based genome-scale metabolic models, only if they have been manually curated and validated with experimental data, and not entirely automatically generated from computational prediction algorithms. Indeed, as described in page 15 of our original manuscript, if a given microorganism or human cell type has a *manually-curated* genome-scale metabolic model, we included the transport reaction of this model into our network; among all the human gut microbes that we examined, ~40 species had manually-curated metabolic models at the time when we initiated our network construction.

In response to the Reviewer's valuable comments, we provide below a figure similar in nature to our original Supplementary Fig. 1. For the vertical axis in the plot, instead of for compounds solely from KEGG, we employed compounds inferred to be from NJS16 using our in-house computational prediction algorithm, which incorporates phylogenetic relationships and KEGG compound annotations for given species. Although this type of figure shall not be used to determine the accuracy of NJS16 because of the same reasons mentioned above, the x and y values now show a stronger correlation ($r = 0.86$ and $P <$

2.0×10^{-5}) than in the original Supplementary Fig. 1 (since this plot still shows a similar pattern, we have omitted its use in our revised manuscript).

Following the Reviewer’s suggestion for a comparison with known growth media information, we provide two additional figures below. The figure on the left shows the distribution of the probability $P(k)$ (vertical axis) that a given microbial species in NJS16 imports k metabolites (horizontal axis) (according to the information provided in NJS16). In comparison, the figure on the right shows the distribution of the probability $P(k)$ (vertical axis) that a given microbial species in NJS16 has k metabolites (horizontal axis) in its defined growth media according to the KOMODO database [*Nat. Commun.* **6**, 8493 (2015)]. We found that the plot in the right figure exhibits peaks at large metabolite numbers (on the horizontal axis) unlike that in the left figure; this could indicate possible false negatives in NJS16’s importable metabolites. However, given KOMODO’s low coverage of NJS16’s species (8.8%), and given that microbes may not necessarily import all the different compounds from their growth media, our results in the right figure may not be entirely suitable to make a fair assessment of NJS16’s coverage and accuracy, and thus warrants a more cautious interpretation. Nevertheless, regarding the Reviewer’s valuable comment, we have included the right figure as Supplementary Fig. 1b in our revised manuscript (Supplementary Fig. 1a is for our original comparison with KEGG compounds); the left figure is included as Supplementary Fig. 2a.

In addition, it is worthwhile to note that, despite several limitations, our work provides a number of experimentally-testable hypotheses, such as a role of B vitamins in the maintenance of T2D-specific microbial communities (page 10). We hence expect that our work will motivate further experimental efforts to validate our predictions.

“2. It is clear that metabolites secreted from given microorganisms are tightly connected to the nutritional resources available at the gut, at least to some extent; there is no reference in the repository to the diet, and to whether the secreted metabolites are related to up-taken metabolites for specific microbes.”

We thank the Reviewer for the valuable comments. Accordingly, we have noted which compounds have dietary origins, such as food macromolecules and vitamins, in our revised Supplementary Data 2. Also, we have discussed the Reviewer’s last comment in our revised manuscript as follows:

(Revised, page page 12) “However, our manual curation approach is not void of drawbacks ... Furthermore, the links between microbes and compounds in our network reflect simple binary information of either the *presence* or *absence* of the corresponding associations ... Substrate-dependent product formation, interdependency of different metabolic pathways, and end-product inhibition of cell growth have yet to be considered ... Our purely connectivity-based network structure should be considered as a map of the *metabolic potential* of the microbial community, rather than of the actual state of metabolism itself.”

“3. In order for this resource to be useful, a reference to known databases IDs should be given both for the organisms (using the NCBI taxonomy tree for example) and for the metabolites (using the KBASE or KEGG repositories). Without this, the resource utility will be quite limited.”

We agree to the Reviewer’s comment. Following the Reviewer’s suggestion, in our revised Supplementary Data 2, we have added the KEGG Compound ID of each metabolite. Given the limited time allowed for our manuscript revision, we have not yet been able to add NCBI taxonomy tree IDs, but we will certainly consider this for our future works.

“4. There is a claim that small molecules (small metabolites) can diffuse through the cell-walls, this point is not addressed in the paper.”

We appreciate the Reviewer’s above comment. When we constructed NJS16, we also included small molecules that diffuse through the cell wall (e.g., H₂ and CO₂, as long as they

serve as primary metabolites for cellular metabolism. In the revised manuscript, we have clarified this point as follows:

(Revised, page 14) “Small metabolites that can diffuse through cell walls (e.g., H₂ and CO₂), thereby not requiring transporter proteins, were also considered, as long as they serve as primary substrates and/or products of cellular metabolism.”

“5. *The selected groups of patients and controls are missing important information needed to reach conclusions regarding metabolic behavior of gut flora (the authors mention that too in the body of the article):*

a. What is the diet of the people in the group? This has a major impact on the structure of the gut flora. I suspect that the diet has much bigger affect on the gut flora, than the one that T2D has.

b. What is the medical treatment that the people in the groups have received? It is known that antibiotics and anti diabetic treatments affect the gut flora (as the authors state). These points should be validated using T2D and control groups with controlled diet & medication.”

These are very excellent points. At the time when we originally analyzed the dataset from Qin *et al.*'s study (2012), information on whether a given microbiome sample was from a metformin-naïve or metformin-treated patient was not publicly available. Now, following the Reviewer's excellent comment, we have looked into Forslund *et al.*'s study, and found that metformin-naïve or metformin-treated information for patient microbiome samples in Qin *et al.*'s study is currently provided. Based on this information, we have *re-performed our entire analysis* with Qin *et al.*'s metformin-naïve T2D samples. In this analysis, metformin-treated samples were excluded in order to avoid medication-confounding effects that the Reviewer pointed out. Among all possible cohorts for both metformin-naïve T2D patients and non-diabetic controls, we selected four cohorts of sufficient and similar sample sizes for parallel analyses (in the original manuscript, we had selected six cohorts, but we have now selected four cohorts because of this criteria of having sufficient sample sizes from metformin-naïve patients). These cohorts are (i) male, young, normal weight (Cohort 1); (ii) male, mid-age, normal weight (Cohort 2); (iii) female, mid-age, normal weight (Cohort 3); and (iv) female, mid-age, overweight (Cohort 4). Full details of the cohort categories are presented in Methods and Supplementary Data 1 of our revised manuscript. With the use of these metformin-naïve samples, our new analysis shows that the main conclusions and insights do not change much from the previous ones, despite some differences in the details of the results. For example, one can compare the following revised Fig. 3 with Fig. 3 of the original manuscript, and can still find remarkable similarities.

Thanks to the Reviewer's above valuable comments, we have thoroughly adopted the new, metformin-free, analysis results in our revised manuscript.

And, as described above, there was no available diet information for our Qin *et al.* samples. In the revised manuscript, we have discussed possible dietary signatures in our samples, as follows:

(Revised, page 7) "As another example, we identified microbial entities in MIN2 and MIN4 that extracellularly degrade triglycerides, thereby releasing diacylglycerols as breakdown products. This result may imply that microbial metabolic activities coupled to host dietary patterns, particularly from high-fat diets (source of triglycerides), contribute to an increased risk in metabolic diseases, including T2D^{33,34}."

(Revised, page 12) "Several limitations of our study should be noted when interpreting our results ... we cannot entirely exclude the possibility of other confounding factors. Notably, the gut microbiota can be significantly altered by one's dietary regimen^{29, 49}, the information of which was not available in the original dataset used in our study. Eventually, replicating our analyses on more finely classified patient cohorts—while

maintaining sufficient sample sizes—could improve control for these potential confounders.”

Additionally, regarding the medication and dietary signatures, it is interesting that Forslund *et al.* noted in their paper [*Nature* **528**, 262 (2015)] that “Suspecting confounding treatments, we tested for influence of diet and antidiabetic medications, finding an effect resulting only from use of metformin.”

“6. As there is already published work on cooperation and competition between species communities done on the level of genome scale metabolic models (GSM) (as can be found in the authors bibliography), a comparison of the results deduced from NJS16 and the ones from the GSMs should be done. A comparison of the results deduced should also be done between NJS16 and the matrices deduced from other GPR databases (KEGG, etc). It is essential to see which have more predictability.”

We appreciate the Reviewer’s careful suggestions. As the Reviewer pointed out, there is a study in which the investigators explored the relationship between (i) the similarity in two species’ nutritional requirements (i.e., competition index) and (ii) the two species’ co-occurrence pattern [*PNAS* **110**, 12804 (2013)]. However, as thoroughly discussed in our answer to one of the Reviewer’s previous questions, we suggest that it is difficult to rely on such a purely bioinformatics-based study in order to determine the accuracy of our literature-curated network (NJS16). As discussed in our previous answer, the “exportable” metabolites suggested by their method show biologically obvious errors, such as glucose 6-phosphate, fructose 6-phosphate, and fructose 1,6-bisphosphate, with too much species-to-species variation in the number of transportable metabolites, even within the same genus. These reasons lead us to question the overall relevance of their results in biological contexts.

Nevertheless, as the Reviewer suggested, we have applied the same measures (i) and (ii) from the aforementioned paper [*PNAS* **110**, 12804 (2013)] onto the Qin *et al.* microbiome samples, while using information on species’ nutritional profiles provided in NJS16. In great agreement with the results of that paper, we have found a mild, yet significantly positive correlation between (i) and (ii) in our own analysis, with $\rho = 0.23$ and $P = 0.03$ (the aforementioned paper found $\rho = 0.21$, although a direct comparison is hard to make because of sample and species differences between the two studies).

Accordingly, we have added this new result to our revised manuscript, as follows:

(Revised, page 4) “[interestingly, our network and microbiome data show that the similarity in two species’ nutritional profiles is positively correlated with the species’ co-occurrence ($\rho = 0.23$ and $P = 0.03$), in agreement with a previous claim²⁰ when the same measures were applied].”

The additional changes made in the revised manuscript are the following:

1. We have improved our manuscript by focusing more on the main message drawn from disease-associated cross-feeding patterns (presented in the last subsection of the Results section). Specifically, this has been emphasized in the abstract and in the Introduction section. In addition, to further highlight the main message, we have made other parts of the manuscript more concise. For example, mathematical formulas in pages 6 and 8 of the original manuscript are now entirely covered by Methods and Supplementary Methods in the revised manuscript. Lastly, the entire text of the manuscript has been thoroughly revised to enhance overall clarity.
2. We have revised our network to improve its quality, mainly by correcting for ambiguous literature information. All the results presented in our revised manuscript have been generated by using this improved version of our network. Supplementary Data 2 provides the details of our improved network, along with a traceable reference list.
3. We have examined the coverage of a species list from Qin *et al.*'s data, as follows:

(Revised, page 14) “For all of our samples, using MetaPhlAn gave rise to a total of 1,219 identified bacterial and archaeal species, which cover ~70% of all the species from unified gut microbiome data with additional data sources (HMP reference genomes, HMSMCP - Shotgun MetaPHlAn Community Profiling, and GutMeta DownLoad Center; accessed August 2016).”

4. In Supplementary Data 2, we have separately marked an extremely “conservative” species list. This list consists of microbial species that have been previously well studied regarding their relationship to the human gut. Supplementary Fig. 3 and Supplementary Table 1 show that our main analysis results do not change much qualitatively between the networks with our full and conservative species lists.

Response to Reviewer #3

We are very thankful for the valuable suggestions and comments offered by the Reviewer. We have carefully considered the Reviewer's comments for the revision of our manuscript. In the following, we discuss the Reviewer's comments and the appropriate changes we made in the manuscript.

“- The starting point of this study is the metagenomic data of the chinese diabetes cohort (qin et al); if the goal of the authors is to provide a general-purpose tool, also the data generated by the HMP and MetaHIT efforts (as well as those of e.g. the appropriate Gordon and Knight lab studies and the recent large-scale population studies) should be used. The species list generated from these data would then be truly representative for the gut diversity and be a sound basis for the network which would be applicable across continents.”

We appreciate the Reviewer's valuable comments. Following the Reviewer's suggestion, we first collected species lists from these microbiome repositories: HMP reference genomes, HMSMCP - Shotgun MetaPhlAn Community Profiling, and GutMeta DownLoad Center (MetaHIT). Next, to examine the extent to which our existing species list (from Qin *et al.*'s study) represents these newly acquired species lists, we combined all, giving rise to a total of 1,783 unique microbial species. Markedly, the Qin *et al.* microbiome data alone have 1,219 species, which cover the majority (~70%) of the total 1,783 species, due to a significant species overlap with the other datasets. Note that ~600 species in our network were almost the maximum species whose reliable metabolic information we could successfully find, out of 1,219 species from Qin *et al.* data. In other words, the network construction itself had been equivalent to being started from ~70% of all species from diverse data sources. Hence, we conclude that our current network covers a substantial portion of gut microbes whose metabolic activities have been characterized to date. Yet, we acknowledge the limited completeness of our network in regards to the aforementioned species coverage. Therefore, we look forward to having our network routinely augmented in the future.

In light of the Reviewer's suggestion, we have revised our manuscript as follows:

(Revised, page 14) “For all of our samples, using MetaPhlAn gave rise to a total of 1,219 identified bacterial and archaeal species, which cover ~70% of all the species from unified gut microbiome data with additional data sources (HMP reference genomes, HMSMCP - Shotgun MetaPhlAn Community Profiling, and GutMeta DownLoad Center; accessed August 2016).”

(Revised, page 12) “Second, our gut microbiota metabolic network is currently limited to bacterial and archaeal species from a particular data source. It needs to be expanded towards other species from different data sources, as well as towards other major phylogenies ...”

“- The reference list contains papers on animal microbiotas, food fermentation etc - it might be that some of these environments contain similar species as those present in the human gut, but its far from certain that the human strains have or perform the same functionality as those from those papers. So despite the obviously impressive amount of work, I worry that the current network is ridden with reactions for which it is uncertain they actually happen leading to noisy and potentially biologically irrelevant results. I suggest the authors focus on human-only reactions to avoid this.

- Similarly, what about oxygen? Some of the bacteria/processes described in the tables are strictly aerobic (acetate production by acetic acid bacteria such as acetobacter). The gut ecosystem is mostly anaerobic so these processes are very unlikely to be important if present at all.

- The authors assign functional capacities from papers (at species level I assume) to metagenome derived phylogenetic annotations. Strain level variation and environmental context can be an important cause of uncertainty here. The authors do not perform any validation of their whole procedure, in silico nor in vitro. For none of the reactions in the network we have an assessment of its reliability, its potential to actually happen in the gut etc - it all comes down to a very greedy, noise sensitive approach, which is likely to result in a network ridden with errors.

- To give just one more example: What is the importance of some of the processes described in the tables for the gut? Ammonia oxidation? Denitrification? Nitrogen fixation? Are they present at all?

- Another one: the authors describe 60 archaea in their list, which is highly unlikely - they are probably found because of mismapping in the taxonomic annotation. I suggest the authors revisit the species mappings of the metagenomic datasets they start from and be extremely conservative (ie more than the original authors); they are more likely to benefit from a smaller, yet more high-quality network than a larger but noisier one.”

These are all very excellent comments. We agree with the Reviewer that our network could possibly include experimental results that may not be straightforwardly translated into *in vivo* events within the human gut, as discussed in our original manuscript (page 13). In line with the Reviewer’s careful suggestion, we therefore repeated our entire analysis for a smaller network constructed upon an extremely “conservative” species list as mentioned by the Reviewer (in contrast to our original, i.e., “full”, species list). This list is composed of microbial species that have been previously well studied regarding their relationship to the

human gut, such as *Roseburia intestinalis*, *Faecalibacterium prausnitzii*, and *Bacteroides thetaiotaomicron*. Our new analysis on this smaller network shows that the main conclusions and insights do not change much qualitatively between our full and conservative networks, despite some differences in the details of the results. More specifically, one can compare the following two figures (left: full network results; right: conservative network results), and can still find remarkable similarities.

Legends are similar to those of Fig. 3 in the revised manuscript.

In Supplementary Data 2 of the revised manuscript, we have clearly marked which species were included in our conservative list. Furthermore, we have added the following to the revised manuscript: Supplementary Fig. 3 and Supplementary Table 1 derived from our analysis on the conservative network, as the counterparts to Fig. 3 and Table 1 derived from the full network, respectively. The following statements have also been added to the revised manuscript.

(Revised, page 4) “(see Supplementary Data 2 for information on all nodes and edges in NJS16, on the associations between macromolecules and their breakdown products, and on which microbial species have been previously well studied regarding their relationship to the human gut).”

(Revised, pages 8–10) “This tendency remains robust in our more conservative analysis (Supplementary Fig. 3c) ... This result is also supported in our more conservative analysis (Supplementary Fig. 3d) ... This tendency did not change in our more conservative analysis (Supplementary Fig. 3e) ...”

In response to one of the Reviewer's subsequent comments, we have extensively revised our network to improve its quality, mainly by correcting for ambiguous literature information, as discussed further below. We are briefly mentioning this here to let the Reviewer know that all the above results derived from the full and conservative networks were generated *after* network revision.

In response to the Reviewer's comment on validating our network, we have performed the following new analysis:

(Revised, page 4) “[interestingly, our network and microbiome data show that the similarity in two species' nutritional profiles is positively correlated with the species' co-occurrence ($\rho = 0.23$ and $P = 0.03$), in agreement with a previous claim²⁰ when the same measures were applied].”

The previous study mentioned above (ref. 20) reported $\rho = 0.21$, although a direct comparison with our own result is not straightforward to make because of some intrinsic differences between the two studies. In addition to the above, our work provides an array of experimentally-testable hypotheses, such as a role of B vitamins in the maintenance of T2D-specific microbial communities (page 10) and the effectiveness of probiotic regimens composed of network influencers, rather than simply of well-known probiotic species (page 8). We hence expect that our work will motivate further experimental efforts to validate these predictions. Interestingly, a recent study [*Nature* **502**, 96 (2013)] is, at least in part, supportive of our finding that pathogenic species are under the metabolic influence of macromolecule-degrading network influencers (page 8).

As previously mentioned, we provide, in Supplementary Data 2 of our revised manuscript, an extremely conservative list of species that have been previously well studied regarding their relationship to the human gut. Analysis on this set of microbes was done to address the Reviewer's concern that results on our full list of species may not be translated well into known *in vivo* gut microbiology. However, we would like to opine that an expanded resource, such as our full list of species, could still provide value as the breadth of knowledge in gut microbiology continues to broaden. Indeed, there have been efforts to revisit some biochemical processes that were once considered to be irrelevant in the gut. For example, as in the Reviewer's comments, oxygen-driven metabolic processes have long been thought to be irrelevant in the gut. Nevertheless, recent findings regarding the presence of an oxygen gradient in gut mucosa have suggested the importance of oxygen for the composition of the gut microbiota, and that even strict anaerobes, such as *Faecalibacterium prausnitzii* and *Bacteroides fragilis*, can surprisingly benefit from small amounts of oxygen as electron acceptors [*ISME J.* **6**, 1578 (2012); *Nature* **427**, 441 (2004); *Gastroenterology* **147**, 1055 (2014)]. Furthermore, regarding microbes whose presence in the gut has been questioned to date (although not of central interest of our study), we introduce the following story of

cyanobacteria: as cyanobacteria are photosynthetic organisms, it has been assumed that their traces from stool simply represent genomic material derived from ingestion of chloroplasts. However, a recent whole genome reconstruction of fecal metagenomes suggests that they belong to a new candidate phylum (non-photosynthetic) sibling to cyanobacteria [*eLife* **2**, e011102 (2013)].

“- Also the quality of the literature study itself worries me: for instance, *B. longum* is mentioned as a hydrogen producer, based on Barcenilla *et al.* 2000. In that paper, *B. longum* is also reported to produce butyrate. Both results are unlikely, but hydrogen was included in the database, butyrate not (although both are obviously related). Why?”

We appreciate the Reviewer’s question. We checked the contents of NJS16 in Supplementary Data 2, and found that our network in fact includes *B. longum* as a producer of both hydrogen and butyrate, unlike the Reviewer’s observation. The Reviewer also questioned how likely both of these metabolites are exported from *B. longum*. Currently, only *B. longum* appears to be a butyrate producer among all *Bifidobacterium* species in our network, and this association is based on Barcenilla *et al.* (2000). It is specifically noted in Barcenilla *et al.* (2000) that *B. longum*, although not expected to produce butyrate, was found to produce small amounts of butyrate. This paper, published in *Appl. Environ. Microbiol.*, has been currently cited >400 times, and we are not aware of any extra information that justifies the exclusion of their results (including the production of both hydrogen and butyrate).

“- Import/export of metabolites is notoriously difficult to determine as transporter specificity can be extremely substrate specific. The authors seem to equate breakdown/production with import/export, yet this is far from always the case. Can the authors provide more detail of the type of evidence they used to determine exchange of metabolites between species?”

Although we sincerely appreciate the Reviewer’s question, we are confused on a few fronts: (i) we are well aware that some transporters are substrate specific and can be difficult to annotate comprehensively. However, our literature-based evidence of import/export was mostly based on *experimental observations* (see Methods), so exchange reactions inferred from mere transporter information were not considered in our study. (ii) Breakdown of a macromolecule, and the ensuing microbial-group-led release of breakdown products into the microenvironment, were *not* considered the same phenomenon in our study, either conceptually or mathematically (Methods and Supplementary Methods). (iii) Literature-based *experimental observations* of metabolite consumption/production are the basis of our network, NJS16. However, we did acknowledge that this information alone does not

guarantee *active cross-feeding*, as stated in our revised manuscript (page 12) that “... our purely connectivity-based network structure should be considered as a map of the *metabolic potential* of the microbial community, rather than the actual state of metabolism itself.”

We welcome any clarifications from the Reviewer to allow us to further address this question.

“- Likewise, there is something called end-product inhibition. Two bacteria producing the same end product limit each other’s growth. A classic example is hydrogen (but not in all cases) - but this is not taken into account in the network. Also, there is a link between substrate fermentation and nature and amount of metabolites produced. By splitting into import and export, this link was cut by the researchers, which I don't think is a good representation of the truth.”

We thank the Reviewer for the excellent comments. While certainly important, such sophisticated details (including end-product inhibition) would be more suitable to be incorporated into dynamical growth models, rather than into our network, whose current version is based solely on network topology. We have discussed these issues in our revised manuscript, as follows:

(Revised, pages 12–13) “However, our manual curation approach is not void of drawbacks: ... Substrate-dependent product formation, interdependency of different metabolic pathways, and end-product inhibition of cell growth have yet to be considered, and these would be better described by constraint-based genome-scale metabolic models ... our network could be utilized to generate computational models ... Promising approaches in this front can be constraint-based methods^{22,24,55-57} and kinetic modeling^{23,58,59}.”

“- With regard to the metformin confounder, did you contact the original authors of the diabetes study to obtain metformin usage data? The diabetes results are unlikely to be very robust without that knowledge”

We appreciate the Reviewer’s valuable comment. At the time when we originally analyzed the dataset from Qin *et al.*’s study (2012), information on whether a given microbiome sample was from a metformin-naïve or metformin-treated patient was not publicly available. Now, following the Reviewer’s excellent comment, we have looked into Forslund *et al.*’s study, and found that metformin-naïve or metformin-treated information for patient microbiome samples in Qin *et al.*’s study is currently provided. Based on this information, we have *re-performed our entire analysis* with Qin *et al.*’s metformin-naïve T2D samples. In this analysis, metformin-treated samples were excluded in order to avoid medication-confounding

effects that the Reviewer pointed out. Among all possible cohorts for both metformin-naïve T2D patients and non-diabetic controls, we selected four cohorts of sufficient and similar sample sizes for parallel analyses (in the original manuscript, we had selected six cohorts, but we have now selected four cohorts because of this criteria of having sufficient sample sizes from metformin-naïve patients). These cohorts are (i) male, young, normal weight (Cohort 1); (ii) male, mid-age, normal weight (Cohort 2); (iii) female, mid-age, normal weight (Cohort 3); and (iv) female, mid-age, overweight (Cohort 4). Full details of the cohort categories are presented in Methods and Supplementary Data 1 of our revised manuscript. With the use of these metformin-naïve samples, our new analysis shows that the main conclusions and insights do not change much from the previous ones, despite some differences in the details of the results. Thanks to the Reviewer's above valuable comments, we have thoroughly adopted the new, metformin-free, analysis results in our revised manuscript.

“- P7-8: Diabetes results - I actually wonder why one needs a metabolic network to find the results described here. Can't these results be found by simple differential pathway analysis?”

If the Reviewer's comment refers specifically to the representative microbe-host interactions on pages 7 and 8 of the original manuscript, we mostly agree to the Reviewer's comment, as those results were largely derived from our analysis of differentially abundant or scarce microbial entities (see Methods and Supplementary Methods). To clarify, this part is presented within the section of microbial metabolic influence networks because these significant microbe-host metabolic interactions complement our network analysis results on microbe-microbe interactions.

“- I'm not sure how to interpret the data in dataset 3 B. What with conflicting signs columns F and G? See also discussion on page 7. E. rectale is reported to compete with M. paludicola for hydrogen gas, but, to my knowledge, E. rectale produces hydrogen. Also, M. paludicola does not seem to appear in dataset 2 B. Why?”

We thank the Reviewer for the careful questions. We believe the Reviewer is referring to columns E and H, and not columns F and G (column F is just the sign of the value in column E, and column G contains metabolite names). The value in column E is the overall metabolic influence (overall W_{ij}) a microbial entity (the source; column C) has on another microbial entity (its target; column D). Here, a positive and negative value denotes the source's positive (i.e., growth promotion) and negative (i.e., growth inhibition) metabolic influence on its target's growth, respectively. Now, the value in column H denotes a metabolite-specific W_{ij} ; in other words, the extent of contribution a particular metabolite has towards a source's overall metabolic influence (column E) on its target. A positive value here means that the

source is most likely cross-feeding that particular metabolite (column G) to its target (i.e., growth-promoting effect); a negative value here means that the source is competing with its target for consumption of that particular metabolite (i.e., growth-inhibiting effect). The sum of all metabolite-specific W_{ij} s (in column H) for a unique source-target pair equals the overall W_{ij} (in column E) for the same source-target pair.

Conflicting signs can appear among values in columns E and H. Let's consider an example where two microbial entities (a source and its target) compete for the consumption of metabolite "A". In this case, the W_{ij} specific to metabolite "A" is negative ($W_{ij_A} < 0$). Continuing with this example, let's assume that the source produces metabolite "B", which happens to be consumed by its target. In this case, the W_{ij} specific to metabolite "B" is positive ($W_{ij_B} > 0$). In completion of this example, let's further assume that metabolites "A" and "B" are the only ones underlying the metabolic influence of the source on its target. The sum of W_{ij_A} and W_{ij_B} , leading to the overall W_{ij} , can be either positive or negative (we ignore for this case a sum of 0). Therefore, from this example, we can see that the sign of W_{ij} (value in column E), and that of either W_{ij_A} or W_{ij_B} (value in column H), can indeed be opposite.

In addition, the Reviewer noted that *M. paludicola* does not appear in Supplementary Data 2b. *M. paludicola* is a methanogen, as shown in Supplementary Data 2d. In Supplementary Data 2b, one can find that methanogens are collectively identified as hydrogen consumers according to ref. 190 and 256 (they were ref. 191 and 257 in the case of the original Supplementary Data 2b).

We completely agree to the Reviewer's correct point that *E. rectale* exports hydrogen, but does not import it. We found that erroneous parts in our original network, including *E. rectale*'s hydrogen consumption, were mostly made by unintended misinterpretations of ambiguous information from the literature. For example, some genus-level information had been loosely generalized for all individual species members. Thanks to the Reviewer's very valuable comment, multiple authors have participated in carefully identifying and correcting these errors, leading to an extensive revision of our original NJS16. This network revision process has led to the correction of a total of 463 links in the network (as one of such corrections, *E. rectale* in our network is no longer a hydrogen importer, but is exclusively a hydrogen exporter, as the Reviewer pointed out). Our revised network now contains 592 microbial species and human cell types metabolically interacting through 4,990 small-molecule transport and macromolecule degradation events.

All the results presented in our revised manuscript have been generated by using this considerably improved version of our network. Supplementary Data 2 now provides the details of our improved network, along with a traceable reference list. Moreover, we have added the following caution for the readers of the manuscript: "However, our manual curation approach is not void of drawbacks: despite our best efforts, the manually-curated network may involve possible misinterpretations of the literature information ... (traceable

literature references are provided in Supplementary Data 2)” (revised, pages 12–13). In our revised manuscript, we found it appropriate to add an additional author (J. J. T. Cabatbat), considering this author’s substantial contribution to the revision and improvement of our network.

“- I would avoid classifying metabolites as 'toxic'. That depends entirely on context and concentration.”

We generally agree to the Reviewer’s viewpoint that toxicity of a given chemical compound can be highly context- or dosage-dependent. However, in our manuscript, we used the word “toxicity” to indicate a metabolite’s *general* tendency of leading to adverse health effects, and we believe that our designations would not cause serious concerns against appreciating the overall trends in Fig. 3d-f. We have also tried a modified classification scheme, wherein some metabolites were classified into an “ambiguous” or “neutral” category; this modified analysis did not lead to much change in the outcomes in Fig. 3d-f. Furthermore, although we acknowledge that the distinction between “toxic” and “non-toxic” does not represent a perfectly ideal dichotomy, we have kept the use of these terminologies in part to avoid otherwise overly and unnecessarily complex analyses and interpretations. In this regard, we provide the following note in our revised manuscript:

(Revised, page 9) “(toxicity of each metabolite was determined based on its general tendency of leading to adverse health effects, although such properties can be highly cell-type- and/or dosage-dependent)”

“- In supplemental Table 2, sheet b, more background on the literature-reported relationships would be helpful to know how reliable the experiment is. For instance, the sheet could at least report the experimental system (animal model, bioreactor, plate, etc.) and in future could report more details, such as medium, pH, temperature, atmosphere, organism grown alone or in a mixture etc., all of which may affect which metabolites are produced and consumed.

- W_{ij} does not weight the importance of metabolites for the survival of species j . The metabolic influence of species i on j will be a different one depending on whether or not a metabolite produced by i can be replaced by another metabolite in i 's absence. Also, species may produce and consume different metabolites depending on the concentration of these metabolites, i.e. they may exhibit different metabolic strategies. The predictions of influencers have therefore to be experimentally validated and should not be presented as final.

- The main network is summarized in supplemental Table 2, sheet f. As a courtesy to readers that want to explore this network, please provide it in a format easily readable

by graph editors (e.g. xgml), with at least one node attribute that provides the node type (metabolite or microorganism) and the consumption/production status and, if possible, the literature source as edge attributes. Additional node attributes could include the lineage for microorganisms and the KEGG identifier for metabolites.”

These are very excellent comments. In regards to the first and last comments requesting a much more in-depth presentation of the literature information, we agree that addressing them will be really helpful to the readers of the manuscript. In our revised Supplementary Data 2, we have added the KEGG Compound ID of each chemical compound in our network. Currently, given the very limited time allowed for our manuscript revision, the best practical strategy for us is to consider the rest of the above valuable suggestions in our future works. In response to the Reviewer’s second comment on the W_{ij} ’s limitations, we have discussed the following points in our revised manuscript (partially overlapped with our previous answer):

(Revised, page 12–13) “Furthermore, the links between microbes and compounds in our network reflect simple binary information of either the *presence* or *absence* of the corresponding associations, whereby the degree of activity of those transport reactions, or individual organisms’ growth requirements, are not yet distinguishable ... these would be better described by constraint-based genome-scale metabolic models. Our purely connectivity-based network structure should be considered as a map of the *metabolic potential* of the microbial community, rather than of the actual state of metabolism itself ... Thus, our work calls for the need to develop high-throughput, quantitative techniques for identifying and validating ... microbial metabolic interactions (... positive/negative metabolic influences) on a global network scale ...”

We expect that the full details of W_{ij} provided in Methods and Supplementary Methods will be helpful for the readers to figure out its current limitations. Note that refining W_{ij} according to all of the Reviewer’s suggestions would require a collection of extensively broader and much more sophisticated information, which is beyond the scope of our study. These suggestions could be more successful in the framework of dynamical metabolic modeling, as described above. Regarding the prediction issue of network influencers in the Reviewer’s comment, as we previously answered, it is interesting to note that a recent study [*Nature* **502**, 96 (2013)] is, at least in part, supportive of our result of pathogenic species being under the metabolic influence of macromolecule-degrading network influencers (page 8).

“- On page 6/17, m is not explained. Is m referring to all species that can produce/consume metabolite k?

- I'm not sure how to interpret the data in dataset 3 B. What with conflicting signs columns F and G? See also discussion on page 7. E. rectale is reported to compete with

M. paludicola for hydrogen gas, but, to my knowledge, *E. rectale* produces hydrogen. Also, *M. paludicola* does not seem to appear in dataset 2 B. Why?"

To the Reviewer's first question, our answer is that m is defined as an index for each microbial entity (like i for each microbial entity), as suggested in our indexing system for n_i and n_m , and $g_{ik}^{p(c)}$ and $g_{mk}^{p(c)}$ in the formula of W_{ij} (pages 6 and 17 of the original manuscript). Because n_m is multiplied by $g_{mk}^{p(c)}$ in the formula of W_{ij} , m is effectively only used for all microbial entities that produce or consume metabolite k , as the Reviewer correctly stated. In cases where details of m were omitted in the index explanations, we did so to avoid unnecessary redundancies in our explanations.

The Reviewer's second question is identical to (or duplicated from) a previous question. Our full answers have already been given to that previous question.

"- "Each species in NJS16 imports 5.2 and exports 3.7 metabolites" - the median is a more meaningful average than the mean here"

We completely agree to the Reviewer's point. Accordingly, we now provide both mean and median values in page 4 of our revised manuscript.

"- Overall the paper could be better written, especially given the broad scope that NCOMMS aims at. Its very dense & complex with a lot of different messages in many directions - overall it could benefit from a major overhaul to increase accessibility for a wider audience."

We appreciate the Reviewer's valuable suggestion. Accordingly, we have improved our manuscript by focusing more on the main message drawn from disease-associated cross-feeding patterns (presented in the last subsection of the Results section). Specifically, this has been emphasized in the abstract and in the Introduction section. In addition, to further highlight the main message, we have made other parts of the manuscript more concise. For example, mathematical formulas in pages 6 and 8 of the original manuscript are now entirely covered by Methods and Supplementary Methods in the revised manuscript. Lastly, the entire text of the manuscript has been thoroughly revised to enhance overall clarity.

The additional changes made in the revised manuscript are the following:

1. In Supplementary Fig. 1b, we have analyzed known growth media information. In addition, in Supplementary Data 2, we have separately marked food macromolecules, vitamins, and other compounds that can be derived from host diet.

2. We have added the following statement to clarify our network construction method:

(Revised, page 14) “Small metabolites that can diffuse through cell walls (e.g., H₂ and CO₂), thereby not requiring transporter proteins, were also considered, as long as they serve as primary substrates and/or products of cellular metabolism.”

Reviewers' comments:

Reviewer #1 (Remarks to the Author):

The revised manuscript is indeed improved though it does not address the many comments around using genome-scale metabolic network. The conclusions regarding, e.g. cross-feeding, thus remain speculative. Similarly, the network-based inference for T2DM is associative and concluding on causality of vitamins as "bad metabolites" is not warranted and may even cause false medical interpretation.

Reviewer #2 (Remarks to the Author):

[No further comments for author.]

Reviewer #3 (Remarks to the Author):

Review of revised manuscript by Sung et al

The authors have performed substantial additional work to improve the quality of the literature-curated network, which I greatly appreciate. I feel that this will now constitute a very nice resource for the gut modeling community. With regard to the network, I only have two additional comments which should be implemented:

- 1) As stated before - as a courtesy to readers that want to explore this network, please provide it in a format easily readable by graph editors (e.g. xgml), with at least one node attribute that provides the node type (metabolite or microorganism) and the consumption/production status and the literature source as edge attributes.
- 2) The authors argue that oxygen has a role in the gut ecosystem. I am willing to accept that, but then oxygen should be added to the reference network.

With regard to the novel mathematical framework, I feel that

- 3) the authors should clarify and justify (in the text) the assumptions they are making: steady state of metabolites, breakdown of macromolecules assumed to be much quicker than explained by sum of degrading species (synergistic), more metabolites generated through macromolecule breakdown than through production by microorganisms etc.
- 4) the authors should clarify and justify why they perform 'additional link filtering' (suppl methods p14)?
- 5) the authors should also make the (documented) code for metabolic influence computation available for the community

The 5 comments above should be easy to implement and I do not have any further comments about that section – the network + mathematical framework is an important and valuable contribution to the field and definitely worth publishing.

However – I unfortunately do still have serious reservations about the T2D section and all the biological conclusions that come with it. The main reason for this lies in the following sentence:

"For the cohorts with statistically enough R's ($R \geq 25$), we used the Benjamini–Hochberg false

discovery rate (FDR) correction for multiple testing ($FDR < 0.4$). "

6) Does this mean that, when searching for species differing in abundance between T2D and control, you only correct for multiple testing if you find more than 25 significantly differing species? Why? MTC should always be performed?

7) Furthermore, and most importantly, the threshold of FDR (< 0.4) is unacceptably high – I have never come across a publication with such a high threshold. The authors should repeat their analysis with $FDR < 0.1$ and only report those results. Currently, they are doing very appealing yet extremely speculative interpretations on very shaky and statistically unsupported results.

Response to Reviewer #1

We are very thankful for the careful comments offered by the Reviewer. We carefully considered the Reviewer's suggestions for the revision of our manuscript. In the following, we discuss the Reviewer's comments and the changes we made in the manuscript.

“The revised manuscript is indeed improved though it does not address the many comments around using genome-scale metabolic network. The conclusions regarding, e.g. cross-feeding, thus remain speculative. Similarly, the network-based inference for T2DM is associative and concluding on causality of vitamins as “bad metabolites” is not warranted and may even cause false medical interpretation.”

We would like to thank the Reviewer for the above valuable comments regarding some speculative conclusions from our study. Following the Reviewer's comments, we have removed sections from our manuscript that could be possibly misleading, particularly those involving our diabetes analysis related to cross-feeding and potential toxicity of vitamins. On the other hand, we have focused more on the description of our gut microbiota metabolic transport network (NJS16), and its application towards community-level reconstruction based on our mathematical framework (for inferring a microbial metabolic influence network) as well as relatively straightforward results and interpretations. For example, to the main text, we have added: a new figure describing the structural properties of NJS16 (Fig. 2), representative mathematical formulas for constructing the metabolic influence network, and another new figure presenting relatively straightforward results from the metabolic influence network (Fig. 4). Accordingly, we believe that our manuscript has been significantly improved thanks to the Reviewer's valuable comments. In line with this revision, we have changed the manuscript title into “Global metabolic interaction network of the human gut microbiota for context-specific community-scale analysis”.

The following additional changes were made in the revised manuscript:

1. In Supplementary Methods, we have provided more detailed explanations for the individual mathematical assumptions that underlie the construction of our metabolic influence network.
2. We have strengthened the value of our network (NJS16) as a useful resource for the scientific community by correcting for remaining ambiguous literature information. To facilitate the broad usage of this network, we provide it in (graph editor accessible) markup language file format (.xml) in our Supplementary Data file. We also provide in another Supplementary Data file a well-documented version of our code that implements the pipeline for the computation of microbe-microbe metabolic influences.
3. We have re-performed our entire analysis with a more stringent FDR cut-off (FDR<0.1). In this new analysis, we have focused only on one cohort (male, mid-age, and normal weight), which has the largest microbiome sample size among all, has comparable sample numbers in both T2D and control groups, and has the most abundant microbial entities passing the new FDR cut-off. The focusing on only one cohort has also prevented a set of speculative biological conclusions, which were previously drawn from the commonality analysis of multiple cohorts.
4. For the microbial organisms in NJS16, we have added information to Supplementary Data 2 regarding their oxygen requirements, according to the Integrated Microbial Genomes & Microbiome Samples (IMG/M) database (<https://img.jgi.doe.gov/cgi-bin/m/main.cgi>).

Response to Reviewer #3

First of all, we are pleased to see that the Reviewer positively appreciates our revised manuscript. The Reviewer stated *“The authors have performed substantial additional work to improve the quality of the literature-curated network, which I greatly appreciate. I feel that this will now constitute a very nice resource for the gut modeling community ... the network + mathematical framework is an important and valuable contribution to the field and definitely worth publishing.”*

We are very thankful for the careful comments and suggestions offered by the Reviewer. We carefully considered the Reviewer’s suggestions for the revision of our manuscript. In the following, we discuss the Reviewer’s comments and the changes we made in the manuscript accordingly.

“1) As stated before - as a courtesy to readers that want to explore this network, please provide it in a format easily readable by graph editors (e.g. xgml), with at least one node attribute that provides the node type (metabolite or microorganism) and the consumption/production status and the literature source as edge attributes.”

This is an excellent suggestion. We believe that the Reviewer’s idea is a helpful way to facilitate the broad usage of our microbial metabolite transport network as a valuable computational resource to the scientific community. Accordingly, we now provide our entire network (NJS16) as a tab-delimited text file and also in (graph editor accessible) markup language file format (.xml) in a Supplementary Data file. Both file formats can be easily implemented in Cytoscape or other network visualization software platforms. The node and edge attributes suggested by the Reviewer can be found in both files.

“2) The authors argue that oxygen has a role in the gut ecosystem. I am willing to accept that, but then oxygen should be added to the reference network.”

For microbial organisms in NJS16, in our revised Supplementary Data 2, we now provide information on their relationships with oxygen (e.g., anaerobic, micro-aerophilic, facultative, and aerobic) according to the classification by the Integrated Microbial Genomes & Microbiome Samples (IMG/M) database (<https://img.jgi.doe.gov/cgi-bin/m/main.cgi>). Due to such varying degrees of oxygen requirements, as well as still largely unexplored nature of oxygen in the gut (recently beginning to be addressed by the scientific community, as reviewed in our previous reply to the Reviewers’ report), we believe that providing this IMG/M-based

information in Supplementary Data 2 should suffice, as opposed to directly incorporating “oxygen consumption” in our reference network. Therefore, we have not incorporated oxygen as an explicitly separate compound into the reference network; rather, we provide the aforementioned IMG/M-based information on oxygen requirements of each species.

“With regard to the novel mathematical framework, I feel that

3) the authors should clarify and justify (in the text) the assumptions they are making: steady state of metabolites, breakdown of macromolecules assumed to be much quicker than explained by sum of degrading species (synergistic), more metabolites generated through macromolecule breakdown than through production by microorganisms etc.

4) the authors should clarify and justify why they perform ‘additional link filtering’ (suppl methods p14)?”

We appreciate the Reviewer’s careful suggestions. We agree that providing more thorough clarifications and justifications for the assumptions used in our mathematical framework would improve our manuscript. The Reviewer’s points mentioned above are discussed in our revised manuscript, as follows (we have cited additional references when deemed necessary. Because our discussions look rather exhaustive, we have added these discussions to Supplementary Methods, rather than to the main text):

(Revised Supplementary Methods, page 4) “Under the steady-state condition, $\frac{dm_k}{dt} = 0$, where m_k is the concentration of metabolite k (this condition should be interpreted as a useful assumption to simplify our analysis, rather than taken strictly as is. As long as m_k has some characteristic value over time, the steady-state condition can be assumed as m_k ’s long-term behavior).”

(Revised Supplementary Methods, page 6) “By assuming the synergistic effect when multiple species in P_k^1 work together to break down a macromolecule [*Appl. Environ. Microbiol.* **55**, 2247 (1989); *Microbiology* **140**, 3407 (1994)], we consider the multiplication, $S_k^{P_1} \prod_{z \in P_k^1} n_z$, rather than the summation of individual species effects ... We further assume $S_k^{P_1} \gg S_k^{P_0}$, indicating that the production rate of metabolite k by macromolecule breakdown is generally much higher than that by microbial export. For example, both macromolecule degradation (such as proteolysis) and microbial export can provide amino acids, yet, the former reflects an active foraging process, while the latter is likely to be

a rarer event that involves temporal dysregulation of amino acid metabolism [*J. Appl. Bacteriol.* **64**, 37 (1988); *Nat. Rev. Microbiol.* **12**, 327 (2014)] ... We further assume that $S_k^{P_m}$ does not have strong m -dependency (given the scarcity of quantitative information on $S_k^{P_m}$, this is the simplest assumption that we can make for our analysis)”

(Revised Supplementary Methods, page 7) “Based on the lowest-order approximation, we assume that the proportion of each species inside G stays almost constant when the total abundance of G becomes changed.”

(Revised Supplementary Methods, page 11) “Additional link filtering: for pairs of microbial entities pertaining to (iii) and (iv), we perform additional link filtering. When link $i \rightarrow j$ is + and link $j \rightarrow i$ is -, and when i and j are differentially abundant in T2D, we remove link $j \rightarrow i$. This is because the putative metabolic influence $j \rightarrow i$ cannot clearly account for entity i 's elevation in T2D; there can be more plausible causes of entity i 's elevation in T2D, such as metabolic influence $i \rightarrow j$, or other (non-metabolic) factors which are beyond our study's scope [*Infect. Immun.* **73**, 3197 (2005); *Proc. Natl. Acad. Sci. USA* **113**, 3639 (2016)].”

“5) the authors should also make the (documented) code for metabolic influence computation available for the community”

We agree to the Reviewer's suggestion to provide, for the scientific community, a well-documented version of the code used in our metabolic influence computation pipeline. This would facilitate the broad usage of our resource. Following the Reviewer's suggestion, we now include all documented scripts for our code and an accompanying set of instructions in a Supplementary Data file.

“I unfortunately do still have serious reservations about the T2D section and all the biological conclusions that come with it. The main reason for this lies in the following sentence: “For the cohorts with statistically enough R 's ($R \geq 25$), we used the Benjamini–Hochberg false discovery rate (FDR) correction for multiple testing ($FDR < 0.4$). ”

6) Does this mean that, when searching for species differing in abundance between T2D and control, you only correct for multiple testing if you find more than 25 significantly differing species? Why? MTC should always be performed?

7) *Furthermore, and most importantly, the threshold of FDR (< 0.4) is unacceptably high – I have never come across a publication with such a high threshold. The authors should repeat their analysis with $FDR < 0.1$ and only report those results. Currently, they are doing very appealing yet extremely speculative interpretations on very shaky and statistically unsupported results.*”

We are very thankful for the Reviewer’s valuable comments. Following the Reviewer’s suggestion, in our revised manuscript, we have *re-performed our entire analysis* with $FDR < 0.1$. In addition, we have removed potentially-misleading biological conclusions from our manuscript, most of which involve our diabetes analysis, including those related to cross-feeding and potential toxicity of vitamins. In this new analysis, we have focused only on one cohort (male, mid-age, and normal weight), which has the largest microbiome sample size among all cohorts, has comparable sample numbers in both T2D and non-diabetic control groups, and has the most abundant microbial entities passing the new FDR cut-off. The focusing on only one cohort has also prevented potentially-speculative biological conclusions, which were previously drawn from the commonality analysis of multiple cohorts.

On the other hand, we have focused more on the description of our gut microbiota metabolic transport network (NJS16), and its application towards community-level reconstruction based on our mathematical framework (for inferring a microbial metabolic influence network) as well as its relatively straightforward analysis results. For example, to the main text, we have added: new figure describing the structural properties of NJS16 (Fig. 2), representative mathematical formulas for constructing the metabolic influence network, and another new figure presenting relatively straightforward results from the metabolic influence network (Fig. 4). In line with this revision, we have changed the manuscript title into “Global metabolic interaction network of the human gut microbiota for context-specific community-scale analysis”. Furthermore, we have strengthened the value of our network as a useful resource for the scientific community by correcting for remaining ambiguous literature information.

As the Reviewer pointed out, we previously applied FDR correction only to the case with $R \geq 25$. We completely agree to the Reviewer’s points that using such an arbitrary choice of R , along with the use of a rather large FDR cut-off, is not convincingly justifiable. Therefore, in this revised manuscript, we have applied $FDR < 0.1$ without adhering to any R , as the Reviewer suggested.

REVIEWERS' COMMENTS:

Reviewer #1 (Remarks to the Author):

The revised manuscript has addressed most of my comments. It is a very nice resource which I think would contribute towards advancing the field of microbiome research. I have only a few remaining comments.

1. Abstract, line 32: I recommend removing "the first". There is a comprehensive model resource recently published by the Thiele lab - a study that also used literature survey. In any case, it is in general better to leave out the novelty claims and let the readers decide.

2. Line 85: "questionable" is not justified here, as the presented approach is also questionable regarding in vivo biology. Consider "possibly incomplete".

3. Line 90: "To address the above concerns": The manuscript does not address these concerns. It helps towards filling the knowledge-gap.

Reviewer #3 (Remarks to the Author):

Overall, I am ok with publication of this paper. They have much improved their ms according to my suggestions. The only thing I still disagree with is the fact that it seems oxygen still isn't part of the network – which I consider important. If they really do not want to add this, they should add this as an explicit limitation described in the text.

Response to Reviewer #1

First of all, we are pleased to see that the Reviewer positively appreciates our revised manuscript. The Reviewer stated *“The revised manuscript has addressed most of my comments. It is a very nice resource which I think would contribute towards advancing the field of microbiome research.”*

We are very thankful for the careful comments and suggestions offered by the Reviewer. In the following, we discuss the Reviewer’s comments and the changes we made in the manuscript accordingly.

“1. Abstract, line 32: I recommend removing "the first". There is a comprehensive model resource recently published by the Thiele lab - a study that also used literature survey. In any case, it is in general better to leave out the novelty claims and let the readers decide.”

We appreciate the Reviewer’s above comment. Following the Reviewer’s suggestion, as well as the Editor’s suggestion made in our manuscript, the term “the first” has been removed from the abstract. In addition, the reference mentioned by the Reviewer has been cited in the Introduction section.

“2. Line 85: "questionable" is not justified here, as the presented approach is also questionable regarding in vivo biology. Consider "possibly incomplete".”

Following the Reviewer’s, as well as the Editor’s suggestion, “questionable” has been changed into “possibly incomplete or inaccurate to some extent”.

“3. Line 90: "To address the above concerns": The manuscript does not address these concerns. It helps towards filling the knowledge-gap.”

Following the Reviewer’s, as well as the Editor’s suggestion, this phrase has been removed from the Introduction section.

Response to Reviewer #3

We are pleased to see that the Reviewer positively appreciates our revised manuscript. The Reviewer stated *“Overall, I am ok with publication of this paper. They have much improved their ms according to my suggestions.”*

In addition, we are very thankful for the careful comment offered by the Reviewer: *“The only thing I still disagree with is the fact that it seems oxygen still isn’t part of the network – which I consider important. If they really do not want to add this, they should add this as an explicit limitation described in the text.”*

In accordance with the Reviewer’s comment, we have added the following two sentences within the Discussion section: *“Relatedly, oxygen-driven metabolic processes have long been thought to be irrelevant in the gut, but recent findings suggest the potential importance of oxygen for the gut microbiota composition⁵⁰⁻⁵². NJS16 does not have oxygen as an explicit metabolic compound, although Supplementary Data 2 provides information on individual species’ relationships with oxygen.”*